

# Efficient multi-objective calibration and uncertainty analysis of distributed snow simulations in rugged alpine terrain

James M. Thornton[1], Gregoire Mariethoz[2], Tristan J. Brauchli[2,3], Philip Brunner[1]

[1] Centre for Hydrogeology and Geothermics, University of Neuchâtel, 2000 Neuchâtel, Switzerland
5   [2] Institute of Earth Surface Dynamics, University of Lausanne, 1015 Lausanne, Switzerland
[3] Centre de recherche sur l'environnement alpin (CREALP), 1950 Sion, Switzerland

*Correspondence to*: James M. Thornton (james.thornton@unine.ch)



**Abstract.** In steep and complex mountainous terrain, robust simulations of snow accumulation and ablation are crucial to a
wide range of applications, especially those related to hydrology and ecology. Whilst new opportunities exist to integrate
high-resolution spatio-temporal observations in the estimation of uncertain parameters in (a.k.a. "calibration" of)
sophisticated, process-rich snow models, they have not yet been fully exploited. Here, with a view towards improving
representations of snow and ultimately meltwater dynamics in rugged topography, a novel approach to the calibration of a
high-resolution energy balance-based snow model that additionally accounts for gravitational snow redistribution is
presented. Several important but uncertain parameters are estimated using an efficient, gradient-based method with respect to
two complementary types of snow observations – snow extent maps derived from Landsat 8 images, and snow water
equivalent (SWE) time-series reconstructed at two contrasting locations. When assessed on a per-pixel basis over 17 days
that together encompass practically the full range of possible snow cover conditions, snow patterns were reproduced with a
mean accuracy of 85 %. The spatial performance metrics obtained compare favourably with those previously reported, whilst
the temporal evolution of SWE at the stations was also satisfactorily simulated. Uncertainty and data worth analyses revealed
that: i) the propensity for model predictions to be erroneous was substantially reduced by the calibration process, ii) pre-
calibration uncertainty was largely associated with two parameters that were introduced to modify the longwave component
of the energy balance, but this uncertainty was greatly diminished by calibration, and iii) the lower elevation SWE time-
series was particularly valuable despite the comparatively small number of observations at this site. Alongside a gridded
snowmelt dataset, commensurate estimates of firn melt, ice melt, liquid precipitation, and potential evapotranspiration were
also produced. Our study demonstrates the growing potential of combining observation technologies and state-of-the-art
inverse approaches to both constrain and quantify the uncertainty associated with simulations of alpine snow dynamics.



## 1. Introduction

Meltwater derived from the seasonal snowpack currently dominates both the annual groundwater recharge and cumulative streamflow of many mid-elevation temperate mountainous catchments (Gleeson and Manning, 2008). At higher elevations, the progressive ablation of firn and glacier ice throughout summer periods represent important additional sources of liquid water to the terrestrial hydrosphere (Schaner et al., 2012). Globally, these snow and ice-derived meltwaters directly sustain millions of people (Pritchard, 2019) and constitute an ecosystem service of enormous value (Sturm et al., 2017). Under

established climatic conditions, freezing levels fall within the elevation range of such catchments for much of a typical year (Meyer et al., 2019). As such, hydrological regimes which are currently heavily influenced by snow and ice are expected to be greatly affected by warming temperatures (Barnett et al., 2005; Viviroli et al., 2011), with summer low flow magnitudes thought to be particularly vulnerable (Dierauer et al., 2018; Jenicek et al., 2016). Indeed, a wealth of evidence attesting the widespread decline of the glaciers and other hydrologically-relevant components of the mountain cryosphere now exists

(Beniston et al., 2018; Bolch et al., 2012; Bormann et al., 2018; Huss et al., 2017; Klein et al., 2016; Vuille et al., 2018), and the consequential impacts on stream discharges are increasingly detectible (Casassa et al., 2009; Lane and Nienow, 2019; Micheletti and Lane, 2016). Accordingly, predictions of the impacts of future climatic change on the quantity and timing of mountain runoff remain in high demand, and the substantial body of literature in which hydrological models are applied to make such predictions continues to be augmented (e.g. Fatichi et al., 2015; Huss and Hock, 2018). Furthermore, since rain-

on-snow events, convective thunderstorms, and more sustained episodes of frontal rainfall all hold hydrological significance in alpine terrain, especially in terms of flood, debris flow, and landslide hazard (Leonarduzzi et al., 2017; Papathoma-Köhle et al., 2011; Rössler et al., 2014), patterns of liquid precipitation also require quantification. The same applies to evaporative losses which – whilst often comparatively overlooked – can also represent an important component of the alpine water balance (Cochand et al., 2019; Herrnegger et al., 2012; Mutzner et al., 2015).


It is increasingly appreciated that after calibrating hydrological models against stream discharge measurements alone, even independent discharge observations can often be reproduced well whilst internal spatial dynamics – including those pertaining to the snowpack – remain poorly represented (Duethmann et al., 2014; Shrestha et al., 2014). Such situations arise due to a lack of information on internal system functioning in discharge data coupled with the considerable freedom

traditional calibration approaches offer. However, a good understanding of, and ability to reproduce computationally, the spatio-temporal dynamics of all potentially important hydrological processes, and thereby obtain the "right answers" via hydrologically plausible mechanisms (Kirchner, 2006), is a prerequisite to the subsequent generation of reliable future predictions. Assessing simulated patterns of model state variables against spatially distributed observations can provide a more stringent test of model capabilities, and is therefore a means by which the internal consistency of hydrological models

can be enhanced. With specific regards to temperate mountain regions, following the early contributions of Grayson, Blöschl and colleagues (Blöschl et al., 1991; Grayson et al., 2002; Grayson and Blöschl, 2001), significant advancements in the



incorporation of spatially distributed snow observations, derived primarily from satellite images, in both model calibration and evaluation have been made. For example, Finger et al. (2011) used Monte Carlo simulations to assess the value of including snow cover images alongside glacier mass balance and discharge measurements in the multi-objective
performance of a distributed hydrological model of the snow and ice-dominated Rhonegletscher catchment in Switzerland, finding a combination of snow cover and discharge data to be optimal. Duethmann et al. (2014) also employed a Monte Carlo approach, this time in an attempt to quantify the relationship between the information content and the number of snow cover images included in the calibration of a model covering several mountainous catchments in Central Asia. In a comparison of two alternative strategies for the simulation of hydrological processes the high-elevation Andes, Ragettli et al.
(2014) likewise incorporated snow cover data, although in this case for purely evaluative purposes. More recently, as part of an investigation into the mechanisms responsible for marked increases in suspended sediment concentrations that were observed in the upper Rhône in the 1980s, Costa et al. (2018) calibrated a simple snow model using distributed snow observations. Finally, an additional group of studies employed distributed snow images not as evaluation criteria but rather as model input (Berezowski et al., 2015; Wulf et al., 2016).


Whilst the aforementioned examples attest to the considerable progress made, many commonly employed modelling approaches are associated with certain limitations that may limit their utility to generate reliable estimates of snow dynamics, and by extension accurate patterns of the arrival of meltwater at the land surface (which is itself a key control on recharge dynamics), in moderately-sized, topographically complex headwater catchments. This is important because detailed, high-
resolution, physically-based hydrological modelling is increasingly being undertaken in such catchments with a view to improving fundamental process understanding (Carroll et al., 2019; Kurylyk and Hayashi, 2017; Penn et al., 2016).

Firstly, despite the now widespread availability of relevant spatial data, spatially lumped (e.g. Wagner et al., 2017) or only partially distributed (Duethmann et al., 2014; Staudinger et al., 2017) hydrological models, which cannot represent spatial
heterogeneity below the scale of the aggregation unit, continue to be frequently applied. Although such lumped models have their uses, incorporating distributed observations with them is complicated; one must either resort to comparisons of catchment (or some other aerially) averaged snow covered area (SCA) (Ragettli et al., 2014) or else somehow reimpose spatial variability afterwards (Parajka and Blöschl, 2008). Moreover, snow patterns assume utmost importance in applications beyond hydrology, such as the prediction of vegetation species distributions (Randin et al., 2015) and winter
tourism (Grünewald et al., 2010).

Secondly, irrespective of spatial discretisation, most previous hydrological modelling studies that have incorporated distributed snow observations relied on snow products from the Moderate Resolution Imaging Spectroradiometer (MODIS) (Clark et al., 2006; Costa et al., 2018; Duethmann et al., 2014; Engel et al., 2017; Ragettli et al., 2014). The 500 m pixel
resolution at which binary (snow or no-snow) and/or snow covered fraction ($f_{SCA}$) data is available (Rittger et al., 2013) is



simply too coarse for certain applications. For instance, both Ragettli et al. (2014) and Hanzer et al. (2016) report difficulties in capturing the complex snow patterns, such as small patches and snow-free ridges, that are commonly observed in rugged terrain using MODIS imagery. Consequently, the information content of these data with respect to relatively fine scale processes that can nevertheless substantially alter the internal hydrological functioning of steep mountain catchments. Much

higher resolution (30 m) snow maps can be derived from Landsat imagery, but thus far have only been applied to model corroboration or evaluation as opposed to calibration. Additionally, with the notable exception of Hanzer et al. (2016), only a small number of images have previously been considered (Bernhardt et al., 2012; Schöber et al., 2010; Warscher et al., 2013).

Thirdly, empirical temperature and other index-based approaches to estimating snow and ice melt rates (Hock, 2003) remain standard (Addor et al., 2014; Etter et al., 2017; Ragettli and Pellicciotti, 2012), despite it having been suggested that they are incapable of satisfactorily reproducing snow dynamics in complex alpine terrain (Warscher et al., 2013). Provided additional meteorological data are available, more sophisticated energy balance-based approaches (both full physics, multiple snow-layer configurations and simplified versions) have been advocated (Magnusson et al., 2015; Meeks et al., 2017). One

particular attraction of distributed energy balance snow models in steep, complex terrain is that they explicitly represent all the fluxes that influence melt, including the pronounced spatio-temporal variability thereof. Another is that by requiring much reduced (if any) calibration, energy balance models are likely to behave more reliably than their simpler counterparts under forcing conditions that exceed the range of historical observations, as would typically be the case in climate change impact assessments (Mas et al., 2018). Indeed, the most sophisticated full physics, multi-layered snow models such as

Alpine3D (Lehning et al., 2006) can now be coupled with simple conceptual hydrological schemes (Gallice et al., 2016) to simulate snow dynamics and runoff across entire catchments. That said, it remains computationally challenging to simulate wind and gravitational redistribution processes at these scales in an explicitly physical fashion (Brauchli et al., 2017; Mott and Lehning, 2010; Musselman et al., 2015). This poses something of a headache because in very steep terrain, for example, accounting for gravitational snow redistribution is indispensable to hydrologically realistic simulations of SWE evolution

and meltwater patterns (Bernhardt et al., 2012; Kerr et al., 2013); *In extremis*, failure to do so can lead to unrealistic simulated "snow towers" (Freudiger et al., 2017). Fortunately, a variety of pragmatic empirical correction methods and algorithms have emerged to account for such processes (Bernhardt et al., 2012; Marshall et al., 2019; Vögeli et al., 2016). Depending on the nature of the catchment and modelling approach in question, these can be applied to facilitate the generation of more hydrologically plausible snowmelt datasets.


A more fundamental challenge in modelling mountain hydrological systems is that the meteorological inputs are often very poorly constrained. Due to wind-induced gauge undercatch, precipitation measurements are generally systematically underestimated (Kochendorfer et al., 2017b; Pan et al., 2016); a bias that is most pronounced with when the precipitation phases is solid (i.e. snow) and with increasing wind speed. Moreover, even in densely instrumented regions like the





European Alps, meteorological station density decreases substantially with elevation (Pepin et al., 2015). Combined with the high degree of variably that characterises mountain meteorology (Mott et al., 2014), this means that even if the original ground measurements could be made perfectly, the resultant interpolated spatial fields would still be associated with considerable uncertainty. Assessments of the error characteristics associated with common instruments, as in the WMO-SPICE project (Kochendorfer et al., 2017a) and extensive investigations into the appropriateness of various possible spatial

interpolation methods (Tobin et al., 2011) have both been pursued to address this challenge. Interestingly, "inverse" methods, whereby distributed models and snow observations are integrated in some fashion to estimate important, uncertain parameters are also beginning to be applied to this challenge. For instance, in a snow model analysis that included both MODIS and Landsat-derived snow observations for evaluation, Engel et al. (2017) reported that modifications to a "snow correction factor" were required to compensate for biased measurements of winter precipitation and thereby improve model

fits. Shrestha et al. (2014) actually calibrated a distributed, multi-layer water and energy balance model (WEB-DHM-S) to minimise the cumulative error in snow cover patterns (again according to MODIS) and discharge simulations. In doing so, an elevation dependent snowfall correction factor was optimised. A particular novelty of this study was that the correspondence between simulated and observed patterns was expressed at the most granular level possible in calibration; that of individual pixels.


Finally, uncertainty quantification should form a central pillar of all environmental modelling exercises. Whilst some previous studies have directly addresses the issue of uncertainty in SWE reconstructions (Franz et al., 2010; He et al., 2011; Meeks et al., 2017; Slater et al., 2013), this has been predominately been undertaken only at discrete stations locations, and not using distributed models. In relation to this, it must be emphasised that as the sophistication, scale, and resolution

forward model sophistication increases, the efficiency of calibration and uncertainty quantification algorithms becomes a critical consideration; despite ever-increasing computational power, "brute force" approaches involving thousands of Monte Carlo simulations still quickly become impractical.

In this context, the present study addresses the following overriding question: To what extent can high-resolution snow

observations inform distributed simulations of snowpack dynamics in rugged alpine terrain? To achieve this,  a novel approach to the generation of high-resolution estimates of meltwater arrival at the land surface in moderately-sized, rugged catchments is proposed. Initially, a fully-distributed energy balance-based snow model that includes gravitational redistribution is established at high spatio-temporal resolution (25 m, hourly). An objective function that incorporates both high-resolution snow cover maps derived from high resolution satellite imagery and reconstructed SWE time-series at two

locations is then developed and minimised using an efficient, iterative calibration algorithm in order to estimate the values of parameters describing important snow-related processes. Via the inclusion of high-resolution distributed snow observations, SWE time-series, and a model that accounts for gravitational redistribution, our approach can be considered an extension of that presented by Shrestha et al. (2014) to steeper, more rugged catchments. To our best knowledge, Landsat-derived snow





cover images have not been previously used in the calibration of any distributed snow models. Additional distinguishing
features of our work are that the spatial fit metrics are computed for a much larger catalogue of images than previous studies,
and that the uncertainty associated with selected key predictions, including the contribution of different parameters and
groups of observations to its reduction, can be elucidated. Through the application of a dynamic glacier model and additional
algorithms, commensurate spatio-temporal datasets of firn and ice melt, liquid precipitation, and potential evapotranspiration
($ET_P$) were also produced. The outputs of such a methodology hold considerable potential for applications concerned with
alpine snow dynamics.

## 2. Study area

Two adjacent headwater catchments in the western Swiss Alps – the Vallon de Nant and the Vallon de La Vare – comprise
the 36.7 km$^2$ study area (Figure 1). The elevational range is considerable, extending from 950 to over 3,050 m above sea
level (a.s.l). Accordingly, slopes are generally extremely steep, and the topography rugged. At the Last Glacial Maximum,
only the highest peaks protruded above the ice (Bini et al., 2009). An array of Quaternary unconsolidated sedimentary
features therefore overly the complex Mesozoic bedrock arrangements. These unconsolidated features have glacial, fluvial,
and mass movement origins, and several of them – especially those in the valley floors – are believed to host aquifers.

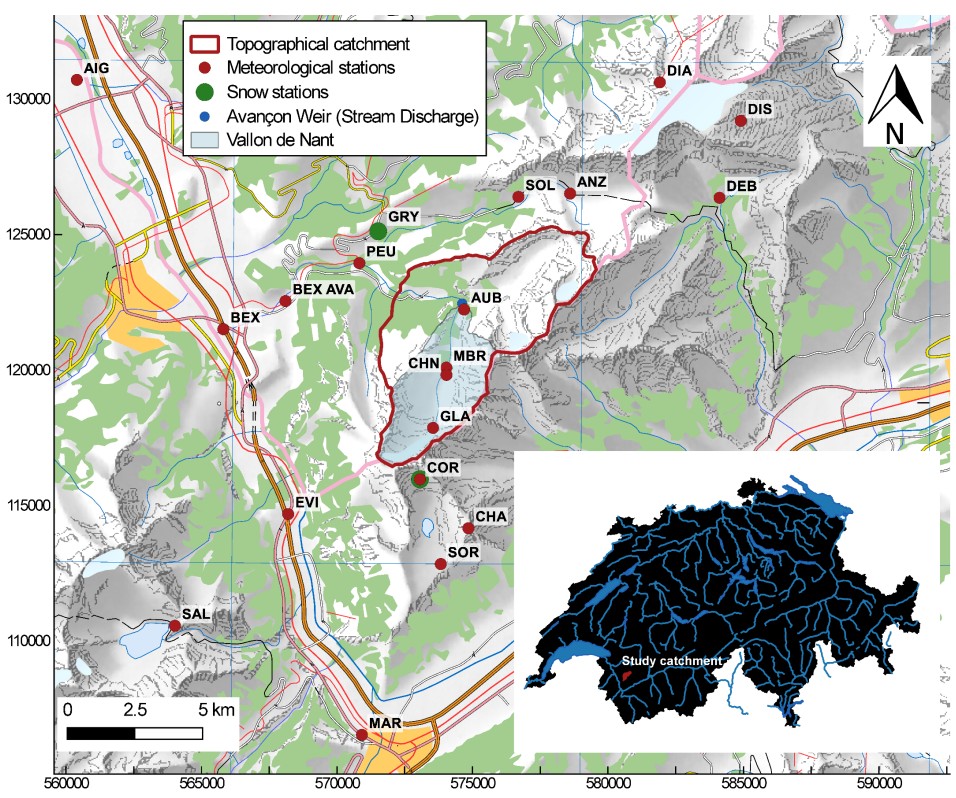

**Figure 1: The situation of the study catchment within Switzerland, and the locations of measurement stations that provided data to the present study. \*The precipitation data at the SOR station were ultimately removed from input dataset as the measured totals seemed too low compared with nearby stations at similar elevations. Background data © swisstopo.**

The area receives abundant annual precipitation, approximately 45% of which falls as snow. Snowmelt is thought to contribute disproportionately to groundwater recharge (Beria et al., 2019). Given the extremely topographic steepness, gravitational redistribution of snow occurs frequently, as evidenced by snow-free slopes and cliffs in the winter months. Intense summer thunderstorms are a further noteworthy feature of the meteorological regime. The surficial hydrology of the Vallon de Nant, which forms the southern section of the study area, is characterised by the eponymous *Nants*, which are temporary torrents whose discharge responds rapidly to rain and snowmelt. Additionally, being shaded by surrounding cliffs, several small glaciers persist at relatively low elevations in the north-facing upper reaches of both sub-catchments.

The Vallon de Nant has been a designated Natural Reserve since 1969, whilst the Vallon de La Vare also remains in an essentially natural state. This makes the study area unusual in the context of the European Alps. Moreover, in terms of its physical characteristics and the processes known to be operating within it, the area can be considered representative of much of the Alps and indeed temperate mountain regions more generally. Reflecting this, the area has formed the site of several





recent investigations into various aspects of the environmental system, including those of Benoit et al. (2018), Grand et al. (2016), Lane et al. (2016), and Vittoz et al. (2009).

## 3. Data

### 3.1. Meteorological forcing

No third-party meteorological datasets with the desired spatial and temporal existed at the outset. The forcing datasets for the present study were therefore developed especially from measurements made at numerous stations (Figure 1). Most of these stations belong to the official networks of MeteoSwiss and the WSL Institute for Snow and Avalanche Research (SLF), and are located outside the study catchment. At these stations, hourly sums for precipitation and hourly means for all other variables spanning the hydrological years 2015—2018 (i.e. 1 October 2014 to 30 September 2018) were downloaded from
the IDAWEB platform (MeteoSwiss, n.d.). Given the relatively low density and distance of many of these stations from the study catchment, coupled with the high spatial variably of precipitation in the area (Brauchli et al., 2018), additional data were obtained from a local network that was installed and is maintained by the University of Lausanne (S. Hiscox, *personal communication*; Michelon et al. (2017) and subsequent updates via personal communication). Several of these stations are located within the study catchment itself. The studies of Wang et al. (2018) is one of many studies that demonstrate the value
of such "local" meteorological data in hydrological modelling. All stations that provided data are listed in Supplementary Table 1. Precipitation data from non-heated gauges were not considered. Also, the precipitation data at SOR were eventually removed from the input dataset because the measured totals were deemed unrepresentative (too low compared with nearby stations at a similar elevations; Brauchli et al. (2018)). Unsurprisingly given the harsh environment and limited access, especially in winter, the local station data were associated with a higher proportion of missing data than those obtained from
the national networks. More intensive processing and quality assurance efforts were also required, most of which was undertaken in R (R Core Team, 2019). The processed time-series were then plotted and inspected interactively using niVis (SLF, n.d.). The hourly time-series themselves are presented in Supplementary Figure 1. The temporal coverage between stations and overall percentage of missing data percentages for each parameter over the four-year simulation period, meanwhile, are shown in Supplementary Figure 2. The simulation period was limited to the four hydrological years 2015-
2018 due to the lack of reliable local meteorological and discharge information prior to this. Although relatively short, the simulation period contains a reasonable diversity of snow conditions, including both relatively snow-rich and snow-poor winters.

As already discussed, obtaining accurate spatial fields of meteorological variables in mountainous regions remains
challenging. This situation has two direct implications for modelling. Firstly, precipitation measurements made using traditional instruments must be corrected for wind-induced undercatch and other factors that cause a systematic bias towards





underestimation. Different corrections were therefore applied depending on the phase of the incident precipitation using Eq. (1):

$$P_{corr} = P\ (snoa \cdot WS +\ snob) \qquad TA\ <\ rstt$$

$$P_{corr} = P\ (liqa \cdot WS +\ liqb) \qquad TA\ \geq\ rstt$$

(1)

where $P$ is measured precipitation (mm), $P_{corr}$ is corrected precipitation (mm), $liqa$ (-) and $liqb$ (-) are global correction
factors for liquid precipitation $snowa$ (-) and $snowb$ (-) are global correction factors for solid precipitation, WS is wind speed (m s$^{-1}$), $TA$ is air temperature (°C), and $rstt$ is the rain-snow threshold temperature (°C).

The rain-snow threshold temperature, here denoted by $rstt$, is not a globally constant deterministic value but instead varies in time and space (Jennings et al., 2018). Accordingly, this parameter was optimised through calibration alongside several
others (see Sect. 4.2). The magnitude of precipitation underestimation in the station measurements is likewise highly uncertain, although the solid precipitating measurements are almost certainly more affected than the liquid precipitation ones. For this reason, both $snob$ and $snoa$ were also subjected to calibration, whilst $liqa$ and $liqb$ were assigned the fixed values of 0.01 and 1.02, respectively. It must be mentioned that neither the solid nor liquid precipitation error characteristics were known at individual stations and/or for individual events, which precluded any more targeted corrections.


The second implication is that careful consideration must be given to the choice of spatial interpolation algorithms. For example, given the pronounced and complex topography, spatial and elevation dependencies in the various meteorological variables should ideally be accounted for. A 25 m resolution digital terrain model (DTM) (Swisstopo, n.d.) was initially established as the model grid. Thereafter, for each variable, an appropriate algorithm was applied to interpolate all available
station measurements of that variable at each time-step. More specifically, to account for their strong elevation dependence, air temperature, wind speed, relative humidity and vapour pressure measurements were interpolated using Elevation Dependant Regression (EDR). In the case of air temperature, the possibility of variability in the linear regression with elevation within a single time-step, including full temperature inversions, was permitted. For precipitation, a linear combination of Inverse Distance Weighting (IDW) and EDR was applied, the ratio of which, $idwedr$, was also subjected to
calibration. In this way, a certain balance between any spatial patterns and (spatially constant) elevational dependence present in the measurements was achieved. Since incoming shortwave radiation and sunshine duration demonstrate more limited elevation dependence, they were interpolated in a straightforward fashion using IDW. Wherever IDW was involved, the maximum search radius was set such that no stations were excluded.





The approach outlined above differs from other studies which have applied (either predefined or calibrated) constant linear temperature-elevation gradient or elevation-precipitation gradients. For example, Brauchli et al. (2017) distributed corrected precipitation measurements from a single station across their study catchment by applying a constant lapse rate of 2 %/100 m. This and other studies (e.g. Naseer et al., 2019) suggest that in complex terrain, such constant lapse rates may be unrealistic. Avoiding such constant gradients can therefore be considered a broadly positive feature of our methodology,

since it ultimately allows more of the information on spatial and temporal structure contained in the local meteorological measurements to be retained. That said, the temporal coverage or "cross-over" between the underlying station data (shown in Supplementary Figure 2) becomes an important consideration. This is because for a given parameter and time-step, only stations returning observations of that parameter at that time-step contribute to the resultant spatial field. In other words, no temporal gap filling or interpolation is undertaken, with each time-step being independent from the last. Consequently, the

uncertainty in the interpolated spatial fields is not constant but rather varies as a function of both the number and location of stations providing measurements for a given parameter at a given time-step.

Finally, corrections were applied to the interpolated hourly temperature and radiation grids to account for topographic shading effects using the scheme of Oke (1987). In this step, an empirical temperature factor, *radc*, was also calibrated. All

of the aforementioned steps were all undertaken using the distributed hydrological model WaSiM (Schulla, 2017).

### 3.2. Snow observations

Two complementary types of data pertaining to observed snow dynamics were developed to constrain the model; i) binary observed snow extent maps derived from Landsat 8 imagery, and ii) SWE time-series at station locations. The spatial imagery provides complete spatial coverage, but only for a number of temporal snapshots and without directly providing

information on the quantity of water stored in the snowpack. Conversely, the time-series provide high-frequency, temporally continuous information on SWE, but only at two discrete locations.

17 cloud-free Landsat 8 scenes falling within the period of meteorological data availability (i.e. the hydrological years 2015—2018) were prepared. For each, the Normalised Difference Snow Index (NDSI; Dozier, 1989) of every pixel was

calculated according to Eq. (2).

$$NDSI_{L8} = \frac{B_3 - B_6}{B_3 + B_6}$$

(2)

*where $B_3$ and $B_6$ are Bands 3 (0.525–0.600 μm) and 6 (1.560–1.660 μm) of a given Landsat 8 (L8) image, respectively.*



Water bodies can have reflectance signatures that produce NDSI values that lie near the boundary between snow and non-snow-covered pixels (having higher and lower NDSI values, respectively). Indeed, the large lakes in our scenes caused significant histogram "spikes" in this range. To prevent these artefacts adversely affecting the subsequent image classification, the lake extents were masked out.

Preliminary snow extent delineations were then made by applying a threshold corresponding approximately to the histogram minima separating the two (usually) distinctive regions, i.e. one mass of pixels with low NDSI values which can be confidently be classified as no-snow, and a second mass of pixels of higher NDSI values which clearly correspond to snow-covered areas. The resultant binary maps were then overlain upon the corresponding True Colour Composite (TCC) images and visual inspections made. Thereafter, bespoke threshold adjustments were manually made to each image in turn to

ultimately reach satisfactory final classifications. In this process, the sensitivity of the derived snow extents to varying the threshold within a plausible range was assessed qualitatively. Finally, to facilitate their integration in the automated calibration process, the maps were re-projected to the CH1903 system (EPSG:21781), clipped to the study catchment, and downscaled from their native 30 m resolution onto the 25 m resolution model grid using the nearest neighbour approach. Figure 2 illustrates the principal phases involved in the generation of these maps, taking 8 April 2015 as an example. The full

catalogue of observed snow extents, which together encompasses practically the full possible range of cover snow conditions, is shown in Figure 3. Supplementary Figure 4, meanwhile, provides comparisons of the TCC images and delineated observed snow extents (alongside their corresponding final simulated outputs).



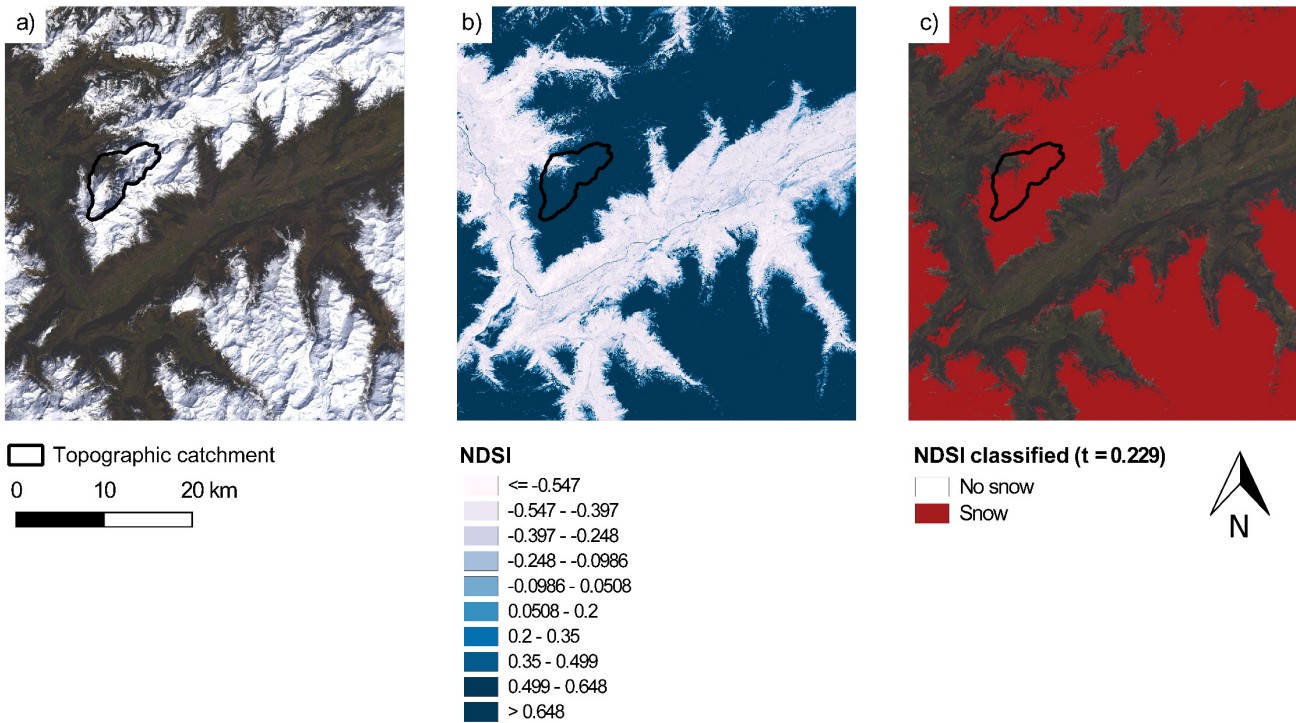

Figure 2: Illustration of the process applied to generate the binary snow extent maps, taking the 8 April 2015 as an example: a) True Colour Composite, b) calculated NDSI raster, and c) final binary observed extent developed by identifying and applying a threshold to (b). In this case, an NDSI threshold of 0.229 was applied. Each image in the observed catalogue was individually inspected and classified in this fashion.



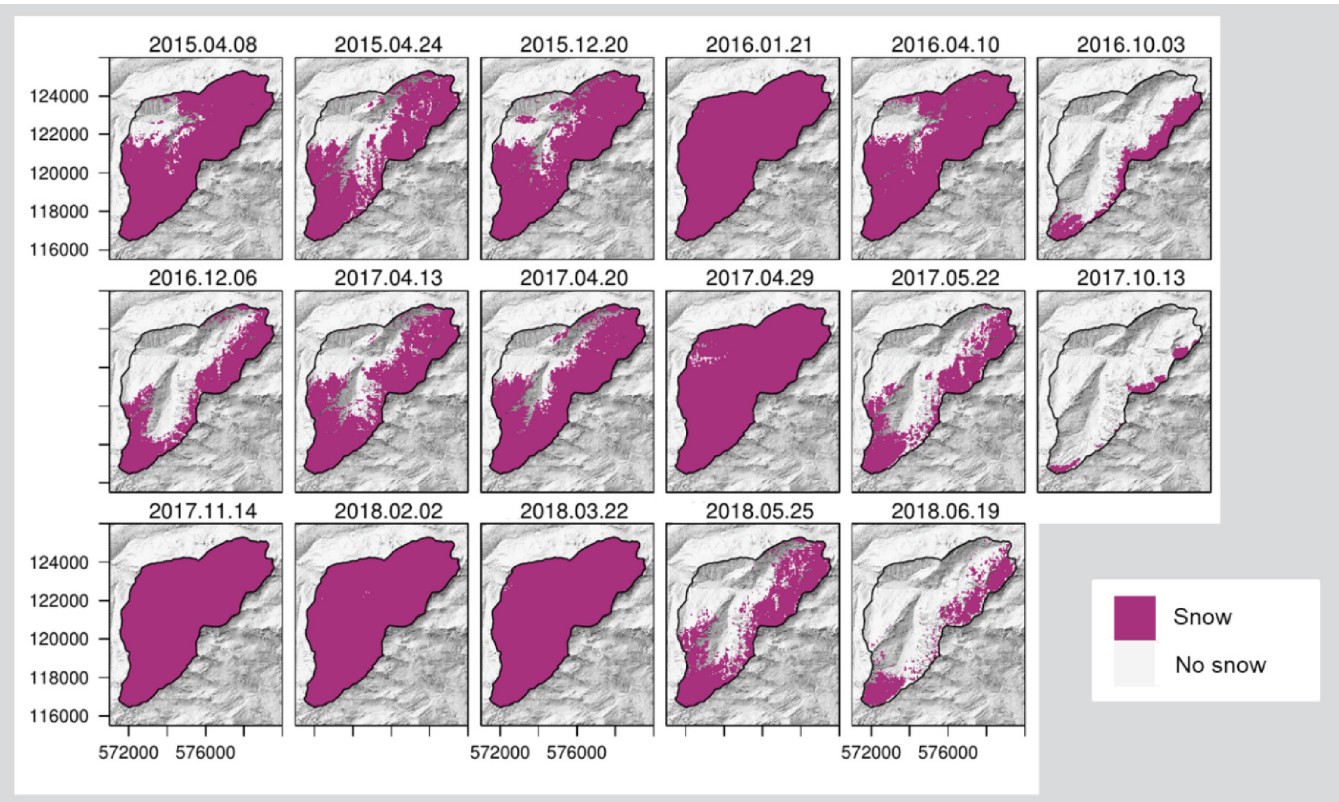

**Figure 3: The complete catalogue of 30-metre resolution observed snow extent maps that was compiled. The date of each image is indicated above the respective pane. Coordinates are in the CH1903 system.**

In order to delineate snow cover extents accurately, it was necessary to apply bespoke thresholds to each image. Appropriate thresholds were found to be somewhat variable from one image to the next, and all were lower than the 0.4 value that has been proposed as standard in the literature. In this regard, our experiences are concordant with those of Härer et al. (2018), who also highlighted the importance of threshold selection. Whilst such an approach remains feasible for a small catalogue of images such as that considered here, it does have implications for the generation of accurate snow maps for a much larger collection of images, which would require an automated procedure.

Besides the more general issues of cloud cover and temporal gaps between images, the identification of snow cover beneath dark forest canopies and areas of shadow (especially due to steep topography) constitute the main challenges when mapping snow extents using the NDSI (Wang et al., 2015). The present study was no exception in this regard. However, whilst it is practically inevitable that a small number of snow-covered pixels under dark forest and in heavily shaded snow-covered terrain will have been misclassified as snow-free, the maps are generally very pleasing in comparison to the TCC images. Moreover, our sensitivity assessment of the classified snow extents to plausible thresholds found it to be quite small (data not



shown). As such, the maps can be considered to represent snow extents with a reasonable degree of accuracy, especially in
the more open upper regions of the catchment where snow patterns are of most interest.

With a view to informing the model more directly with respect to snow water storage, reconstructed time-series of SWE at
two contrasting locations were also developed. (Given the single layer snow model configuration employed only provided
total snow water storage as opposed to fuller information on snowpack density, depth and SWE interrelations, SWE
observations were also required for the practical reason of being comparable with the model outputs). Fortunately, regular
snow measurements were available over the period in question at two stations situated just outside the study catchment, each
lying towards one extreme of its elevational range (see Figure 1). These data were again obtained from IDAWEB. The high
elevation station, Grand Cor (COR; Elevation: 2,602 m), belongs to the SLF's IMIS network (SLF, n.d.). These stations do
not measure solid precipitation directly, but instead record snow height and several other variables that can be used to drive
the 1D physically-based, multi-layer model SNOWPACK (Lehning et al., 2002) (which also underpins the distributed model
Alpine3D). In this way, an hourly time-series of the evolution of SWE at COR over the entire four year simulation period
was constructed.

The second, lower elevation station is located in Gryon (GRY; Elevation: 1,146 m). In contrast to at COR, only (manual)
daily snow height measurements are made here. This prevents the application of SNOWPACK. To generate model
comparable observations nevertheless, the statistical model of Jonas et al. (2009) was applied. This empirical model, which
was constructed from a large sample of snow observations from the Swiss Alps, enables the estimation of snow density as a
function of month of the year and geographical region. In the present application, the parameters corresponding to "Region
1", within which our catchment is located, were taken. The resultant densities were then multiplied by measured snow
heights to give SWE at daily intervals.

As such, the neither of the "observed" SWE time-series, which are presented in Figure 5 (alongside their simulated
counterparts from the final model), are actually direct measurements. Because both locations fall outside the study
catchment, in order to utilise these data model calibration it was necessary to expand the simulation domain slightly.

### 3.3. Stream discharge

A gauging station in the form of a regular concrete weir is located at the outlet of the Vallon de Nant sub-catchment (E:
574,620, N: 122,462). In conjunction with a rating curve that was developed by salt dilution gauging (Ceperley et al., 2018),
this station has provided high-frequency streamflow estimates since spring 2016. Despite the regular cross-section,
considerable uncertainties are associated with these data, especially at flow extremes. These uncertainties are the result of


shifting channel configurations immediately upstream of the weir (A. Michelon, *personal communication*), as well as flows that exceed the range of measurements used in rating curve construction.

## 4. Methods

### 4.1. Simulating snow accumulation, redistribution, and melt

As for the correction of precipitation and interpolation of the meteorological data, WaSiM (Schulla, 2017) was selected to form the foundation of our snow modelling approach. This decision was made on account of its strong capabilities following a thorough review that included testing of possible alternatives. In summary, snow accumulation, redistribution, and melt were calculated at an hourly time-step on the 25 m model grid. First, the precipitation phase was estimated for each pixel and time-step by according to the interpolated air temperature and a transitional range within which both solid and precipitation

can occur (Equation 3).

$$S_{frac} = \frac{rstt + T_{trans} - TA}{2 \cdot T_{trans}} \qquad for \quad (rstt - T_{trans}) < TA < (rstt + T_{trans})$$

(3)

*where $S_{frac}$ is the fraction of the total precipitation that is snow (0-1), TA is the air temperature (°C), rstt is the rain-snow*

*threshold temperature, and $T_{trans}$ is half of the rain-snow transition temperature range (°C).*

$T_{trans}$ was fixed to 1°C (i.e. the total transition range was 2°C), whilst (rstt) took the same (calibrated) value as that applied to distinguish precipitation phase in the earlier correction phase (Eq. 1). Snowmelt was then calculated by solving the surface energy balance for the energy available for melt following the approach of Warscher et al. (2013). In this scheme, the

snowpack is treated as a single homogenous layer beneath the surface, for which the energy balance is computed using Equation (4).

$$Q + H + E + A + G + M_{ae} = 0$$

(4)

*where Q is the shortwave and longwave radiation balance, H is the sensible heat flux, E is the latent heat flux, A is the advective energy supplied by solid or liquid precipitation, G is the soil heat flux (which is small compared to other fluxes and was set here to 2 W m⁻², and $M_{ae}$ is the energy potentially available for melting during a given time-step.*





Melting and non-melting conditions were again distinguished according to *rstt*. When the energy balance is positive (i.e. $M_{ae}$
> 0) and air temperature favourable, melt (*M*) can occur. Finally, *M* is expressed in mm of water by introducing the latent
heat of fusion, $c_i$ (Equation 5).

$$M = \frac{M_{ae} \cdot dt}{c_i}$$

(5)

Sublimation, which can amount to an important component of the alpine water balance (Strasser et al., 2008), is explicitly
modelled in this approach. Two additional scaling parameters, *lwin* and *lwout,* could be modified from their default value of
1 in order to fine tune the incoming and outgoing longwave components of the energy balance, respectively. In this way,
potential errors in both albedo and cloudiness could be accounted for. Both were subjected to calibration, albeit within
relatively strict bounds (see Table 1).


Additionally, gravitational redistribution was simulated using the mass-conservative algorithm implemented by Warscher et
al. (2013) that depends on a topographic analysis and the available mass input. Several steps are involved. Along with the
previously summarised algorithms, these are comprehensively described by both Warscher et al. (2013) and Schulla (2017).
Here, merely the main parameters are discussed. Two parameters represent the critical local slope limits; *mids* controls the
lower inclination limit for gravitational slides, whilst *mads* is the upper inclination angle above which all incoming snow is
immediately transported downslope. Since these slope angles are dependent on the scale of the model grid, they cannot
easily be transferred from previous studies, and were therefore included as calibration targets. Following the advice of
Schulla (2017), two further parameters related to the gravitational redistribution were also calibrated. *frss* is the fraction of
the snowpack at a given time-step that can form a slide. Its value should usually be set to some small fraction of the current
snow storage in a cell, typically on the order of 1% (although this is time-step dependent). *scmd* is an upper depositional
mass limit (mm) for snow flows. Whilst such an approach is capable of estimating plausible snow distribution patterns, it
must be emphasised the specific timing of avalanches cannot be predicted (Warscher et al., 2013).

A simple algorithm designed to account for the redistribution of snow by wind was also proposed by Warscher et al. (2013).
More recently still, algorithms seeking to better represent the interaction of snow and coniferous forest canopies have been
published (Förster et al., 2018). However, neither of these sets of algorithms were included in our final model; the inclusion
of the wind algorithm actually resulted in poorer model fits to the observations, whilst the WaSiM release with the
extensions of Förster et al. (2018) (i.e. version 10.04.01, released on 6 March 2019) came at too late a stage in this work for
them to be thoroughly considered.



## 4.2. Multi-objective calibration

Our model comprises 11 free parameters, which are listed along with the upper and lower bounds that were assigned to each according to prior knowledge, in Table 1. The final estimated values are also indicated to prevent the later duplication of a similar table.

| Parameter | Description | Lower Bound | Upper Bound | Estimated value |
|-----------|-------------|-------------|-------------|-----------------|
| *rstt* | Snow-rain temperature threshold (°C) | 0.0000001 | 3.5 | **0.0266** |
| *snoa* | Snow precipitation correction (-) | 0.0000001 | 0.15 | **0.0283** |
| *snob* | Snow precipitation correction (-) | 1.0 | 1.45 | **1.4500** |
| *radc* | Factor for temperature correction *radc*·(-1.6 .... +1.6) in the radiation correction module | 0.1 | 8.0 | **0.1731** |
| *mads* | Maximum slope for snow deposition (°) | 45.0 | 75.0 | **73.5269** |
| *scmd* | Upper deposition limit for gravitational redistribution (mm) | 0.0000001 | 10.0 | **1.1497** |
| *mids* | Minimum slope for gravitational slides (°) | 20.0 | 48.0 | **43.3811** |
| *frss* | Fraction of snowpack that forms the slide (0-1) | 0.001 | 0.05 | **0.0076** |
| *idwedr* | Relative weight of IDW to EDR in the interpolation of precipitation (0-1) | 0.05 | 0.85 | **0.05** |
| *lwin* | Correction factor for incoming long wave radiation for fine tuning the energy balance (accounting for errors in cloudiness and albedo) | 0.7 | 1.3 | **1.2167** |
| *lwou* | Correction factor for outgoing long wave radiation for fine tuning the energy balance (accounting for errors in cloudiness and albedo) | 0.7 | 1.3 | **1.1920** |

**Table 1: Parameters of the WaSiM model that were subject to calibration. The final estimated parameter values are also reported here to prevent the later duplication of this table.**





A novel, multi-objective calibration approach that incorporated both the spatial snow extents and the reconstructed SWE time-series was developed. For each of the 17 days on which an observed extent map had been produced, and for every model iteration, the spatial component of the overall goodness-of-fit was quantified via the following process:


1. The simulated SWE maps at the end of the days corresponding to observed maps were extracted and clipped to the study catchment extent.

2. Pixels in the simulated SWE maps were reclassified to either snow or no-snow using a 5 mm exceedance threshold (i.e. pixels with SWE > 5 mm were classified as snow covered).

3. According to whether the presence of snow had been correctly simulated in comparison with the observed snow extent maps, each pixel was binned into one of the quadrants of the contingency matrix shown in Table 2.

4. Three related performance metrics were calculated after Aronica et al. (2002) using Equations 6-8.


|  | Observed snow | Observed no snow |
|---|---|---|
| **Simulated snow** | $a$ | $b$ |
| **Simulated no snow** | $c$ | $d$ |

**Table 2: Contingency matrix used for the classification of pixels in a given iteration of model simulations.**

$$F_1 = \frac{\sum_{i=1}^{n} a + \sum_{i=1}^{n} d}{n}$$

(6)

$$F_2 = \frac{\sum_{i=1}^{n} a}{\sum_{i=1}^{n} a + \sum_{i=1}^{n} b + \sum_{i=1}^{n} c}$$

(7)

$$F_3 = \frac{\sum_{i=1}^{n} a - \sum_{i=1}^{n} b}{\sum_{i=1}^{n} a + \sum_{i=1}^{n} b + \sum_{i=1}^{n} c}$$

(8)

*where a, b, c,* and *d* are the quadrants of the contingency matrix (Table 2) and *n* is the total number of pixels.

$F_1$ (Eq. 6) corresponds to the overall proportion of pixels that were correctly simulated. $F_2$ and $F_3$ (Eqs. 7 and 8) typically result in lower scores since they expressly discount pixels that are snow free in both simulations and observations; this
category is often heavily populated because on many days there are typically a large number of observed snow-free pixels at



lower elevations which can generally be reproduced with relative ease (Warscher et al., 2013). For each statistic, a perfect fit between the simulations and observations corresponds to a result of 1. Therefore, following each iteration of the model, the squared residual between each calculated F-statistic and 1 was calculated. Model performance with respect to the observed SWE time-series was quantified according to the squared residuals of simulated and observed values at each time-step.


In constructing a single multi-objective function that could be minimised and thus produce the best overall fit according to both observation types, the subjective process of assigning weights to each individual squared residual had to be addressed. The aim was to ensure that each observation had a certain "visibility" in the calibration process, whilst an appropriate balance between the two main data types was also achieved – considering their relative strengths and weaknesses – was

achieved. The upshot was the hierarchical weighting scheme illustrated in Figure 4. The spatial data were weighted slightly more than the time-series data (60:40). For each day on which snow map comparisons were possible, a double weighting was given to the $F_2$ and $F_3$ values relative to $F_1$ to reflect their more stringent nature. Finally, to account for the fact that snow measurements were made hourly at COR but only daily at GRY, the latter were assigned weights 24 times higher than the former to ensure parity between the stations.

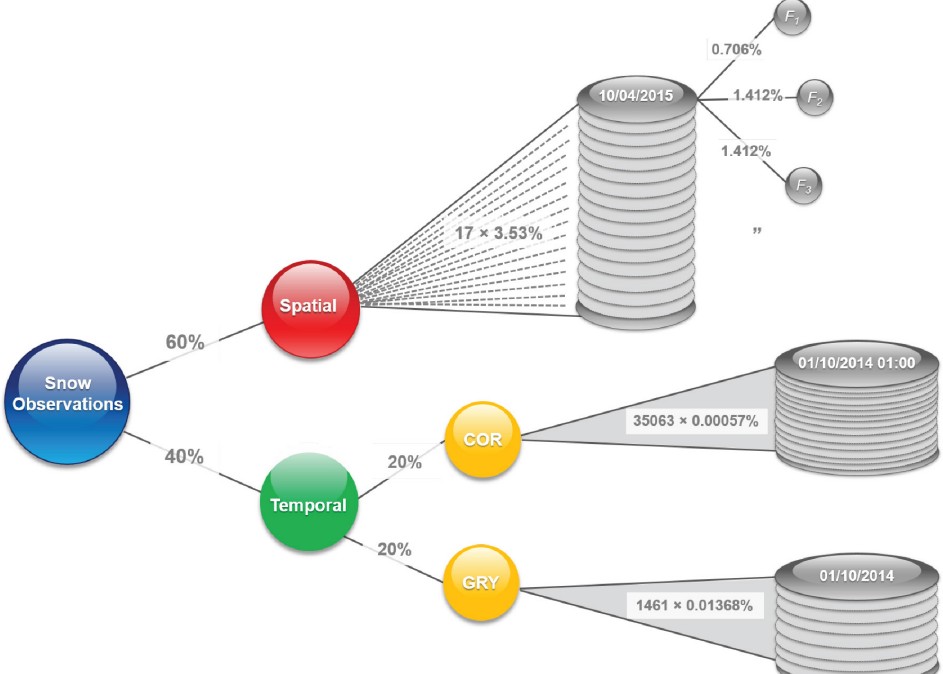


**Figure 4: Schematic diagram illustrating the weights, expressed as percentages (%), that were assigned to each individual snow "observation". For each observed snow extent map, the three F-statistics (with each given a value of 1) were treated as individual observations. In total, there were thus 51 (17 × 3) distinct spatial "observations", 35,063 (hourly) SWE observations spanning the four-year simulation period at COR, and 1,461 (daily) observations spanning the same period at GRY. The 5 observation groups**
**(i.e. $F_1$, $F_2$, $F_3$, COR and GRY) also formed the categories in the linear uncertainty analysis.**

The objective function (*OF*) is represented using mathematical notation in Equation (9).



$$OF = \sum_{i=1}^{17}[wF_1(1 - F_1)^2] \;+\; \sum_{i=1}^{17}[wF_2(1 - F_2)^2] \;+\; \sum_{i=1}^{17}[wF_3(1 - F_3)^2]$$

$$+ \sum_{i=1}^{35063}\left[wCOR(COR\_SWE_{sim} - COR\_SWE_{obs})^2\right] + \sum_{i=1}^{1461}\left[wGRY(GRY\_SWE_{sim} - GRY\_SWE_{obs})^2\right]$$

(9)

*where $wF_1$, $wF_2$, $wF_3$, $wCOR$, and $wGRY$ are the relative weights that were assigned to each observation belonging to the different observation groups according to Figure 4, i.e. 0.706, 1.412, 1.412, 0.0057 and 0.01368 %, respectively, $F_1$, $F_2$, and $F_3$ are the fit statistics calculated for each of the 17 pairs of images according to Eqs (6-8), and $COR\_SWE_{sim}$, $COR\_SWE_{obs}$, $GRY\_SWE_{sim}$, and $GRY\_SWE_{obs}$ are the observed and simulated SWE values at each time time-step at the COR and GRY stations respectively.*

The WaSiM model was then coupled with PEST (Doherty, 2019) – a model-independent, gradient-based parameter estimation tool which uses the Levenberg–Marquardt algorithm. PEST repeatedly runs the model altering the calibration parameter values in each iteration in an attempt to minimise the objective function. PEST was selected primarily due to its efficiency, which is considerably higher than the more commonly applied Monte Carlo-oriented approaches. Indeed, parameter search efficiency was crucial given the relatively high computational demands of our energy balance-based snow 515 model. In practical terms, the coupling was achieved by implementing routines to extract the spatial and temporal model outputs corresponding to the observations and calculate the required statistics in an R script (see Thornton et al., 2019). The final parameters values resulting from this process are presented in Table 1.

### 4.3. Predictive uncertainty and data worth analyses

Following calibration, a linear analysis was conducted to quantify the pre- and post-calibration uncertainty associated with 520 selected individual "predictions" (this term is not used here in a future sense) of interest; namely, the SWE at COR on 1 April in each of the 4 simulated hydrological years (2015-2018), and the $F_1$ snow map metric for the 22 May 2017. In this way, the reduction in uncertainty (if any) attained via the calibration process could be evaluated. The 1 April SWE at station locations is an indicator that is commonly employed by environmental managers in snowmelt dependent regions to assess probable water availability throughout the subsequent summer. Therefore, uncertainties associated with such estimates 525 stemming purely from the model would be of interest if, for example, it was forced using long-terms meteorological forecasts to generate genuine future predictions. The spatial prediction was included because few previous studies have specifically considered uncertainty in predictions of snow patterns. Thereafter, analyses addressing the contribution of individual parameters to pre and post-calibration uncertainty variance, and the information provided by the five observation groups to the calibration process (i.e. "data worth"), were also undertaken. To achieve all of these tasks, tools belonging to



PEST's GENLINPRED suite were applied. For a through description, readers are referred to Doherty (2010, 2019). In contrast to the calibration of the model itself, an identical weight was assigned to all non-zero weighted observations for the analysis (zero-weighed observations being the predictions of interest). Following the advice of Doherty (2010), this weight was estimated by taking the number of non-zero weighted observations (which in this case was 36,570), finding its square root, and dividing the result by the objective function of the calibrated model (91,150), giving a value of 0.002098.

**4.4. Glacier melt, liquid precipitation, and potential evapotranspiration**

To generate the full range inputs for any subsequent distributed hydrological modelling, four additional datasets – liquid precipitation, firn melt, ice melt, and $ET_P$ – were also developed using the WaSiM model. Firstly, to incorporate liquid precipitation, grids referred to as "snowcover outflow" were written at each time-step. These grids represent the combination of any liquid precipitation in the case that a given pixel is snow free, and any snowmelt that arrives at the surface where a

given pixel is snow covered (as calculated by the snow model). As explained in Sect. 3.1, modest fixed corrections were applied to the original liquid precipitation measurements to account for undercatch. Accordingly, in the case of snow-free pixels, the "snowcover outflow" values corresponds to any (corrected, interpolated) rainfall that occurs.

The aerial proportion of glaciers in our study catchment is small (<3 %). The glaciers therefore make a much smaller

contribution to annual, catchment averaged meltwater than the snowpack at this site. Nevertheless, they are responsible for considerable localised meltwater generation in summer. As such, to account for the accumulation, dynamics, and ablation of the glaciers, a dynamic glacier model with radiation correction was also established in WaSiM. The parameters of this model were not calibrated; a decision that may be justified due the overall dominance of snowmelt as well as a lack of glacier-related data; the Glacier des Martinets, for instance, has not actively monitored since 1975 (GLAMOS, 1881-2018). To

simulate the evolution of snow cover on the glacier, an approach identical to in the main snow model was applied. In this way, distinct hourly, 25 m grids of snowmelt, firn melt, and ice melt from glacierised areas were returned. A further benefit of including glacial melt is that it increases the generalisability of our workflow; it is envisaged that with only slight modification (possibly related to the distinction between snow and bare glacier ice in NDSI images), it may be applied to catchments with much higher glacierised proportions. The final WaSiM control file is provided in the associated data and

code (Thornton et al., 2019). In addition to containing all the optimised snow-related parameters, this file indicates the values of the (fixed) parameters that were applied in generating these additional datasets.

In order to eventually sum "snowcover outflow" with the meltwater outputs from the glacier model and thus generate a catalogue of rasters representing the combination of all liquid water arriving at the surface for each time-step (a.k.a. "surface

water input"), some post-processing was required. More specifically, it was necessary to normalise the glacier model outputs according to the glacier covered fraction of each cell at each time-step (since the glaciers were dynamic), which ranged between 0 and 1. Having done this, the "snowcover outflow" grids were summed together with the normalised "snowmelt on



glacier", "firn melt", and "ice melt" for each time-step. These calculations were carried out by executing GDAL scrips in batch via OSGeo4W (GDAL, n.d.; OSGeo4W, n.d.).


Finally, the Penman–Monteith method was used to estimate $ET_P$, again on the 25 m resolution model grid on an hourly time-step. The land cover map that was developed from existing swisstopo datasets (swisstopo, n.d.) and then attributed with appropriate physical parameters to contribute to this estimation is presented in Supplementary Figure 3. Fuhrer and Jasper (2012) provide a fuller example of the application of WaSiM to this end. No additional processing of the $ET_P$ grids was

required. In being of identical spatial and temporal resolution to the snowmelt grids, and moreover having been generated using predominantly physically-based approaches, all resultant datasets can be considered broadly commensurate with one another.

In Figure 6, daily mean "surface water input" and $ET_P$ values that were derived from the hourly outputs are presented on a

monthly basis for the two most recent hydrological years of the simulation period. To further evaluate the hydrological plausibility of our simulations (but without recourse to a full hydrological model), the simulated catchment averaged "snowcover outflow" dataset for the Vallon de Nant sub-catchment was compared with normalised observed discharge at the discharge station (see Figure 7).

## 5. Results

### 5.1. Spatial and temporal model fits

The spatial goodness-of-fit statistics (i.e. $F_1$, $F_2$, and $F_3$) obtained following calibration are shown in Table 3. In general, the F-statistics are reasonably high; across the 17 days, the average of the percentage of correctly simulated pixels was 85 %. As expected, the scores decline progressively from $F_1$ to $F_3$ on every day. Moreover, a relatively high degree of variably between days and seasons was observed, with completely snow covered mid-winter days naturally achieving the best scores.

The lowest $F_1$ value was returned for the 29 April 2017, when a late season snowstorm that briefly blanketed the catchment was unfortunately missed by the model. The corresponding observed and simulated snow maps from which these statistics were produced are shown in Supplementary Figure 4.






| Date | $F_1$ | $F_2$ | $F_3$ |
|---|---|---|---|
| 08/04/2015 | 0.852 | 0.847 | 0.696 |
| 24/04/2015 | 0.786 | 0.698 | 0.511 |
| 20/12/2015 | 0.838 | 0.798 | 0.699 |
| 21/01/2016 | 0.999 | 0.999 | 0.999 |
| 10/04/2016 | 0.871 | 0.849 | 0.819 |
| 03/10/2016 | 0.910 | 0.621 | 0.399 |
| 06/12/2016 | 0.750 | 0.596 | 0.199 |
| 13/04/2017 | 0.744 | 0.569 | 0.504 |
| 20/04/2017 | 0.790 | 0.752 | 0.508 |
| 29/04/2017 | 0.567 | 0.561 | 0.561 |
| 22/05/2017 | 0.846 | 0.626 | 0.434 |
| 13/10/2017 | 0.958 | 0.131 | 0.073 |
| 14/11/2017 | 1.000 | 1.000 | 1.000 |
| 02/02/2018 | 0.997 | 0.997 | 0.996 |
| 22/03/2018 | 0.963 | 0.963 | 0.963 |
| 25/05/2018 | 0.773 | 0.584 | 0.348 |
| 19/06/2018 | 0.844 | 0.328 | 0.089 |
| *mean* | **0.852** | **0.701** | **0.576** |

**Table 3: The post-calibration F-statistics that quantify spatial goodness-of-fit for each of the 17 days.**

Figure 5 shows the comparison between simulated and observed SWE at the two measurement stations.



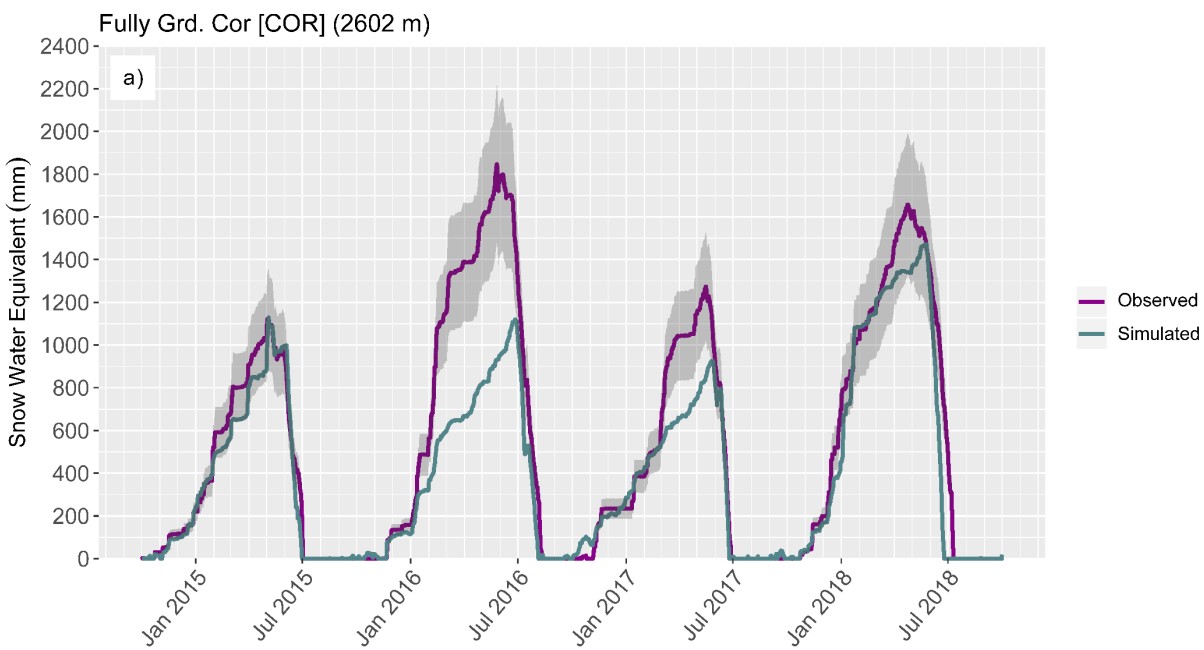

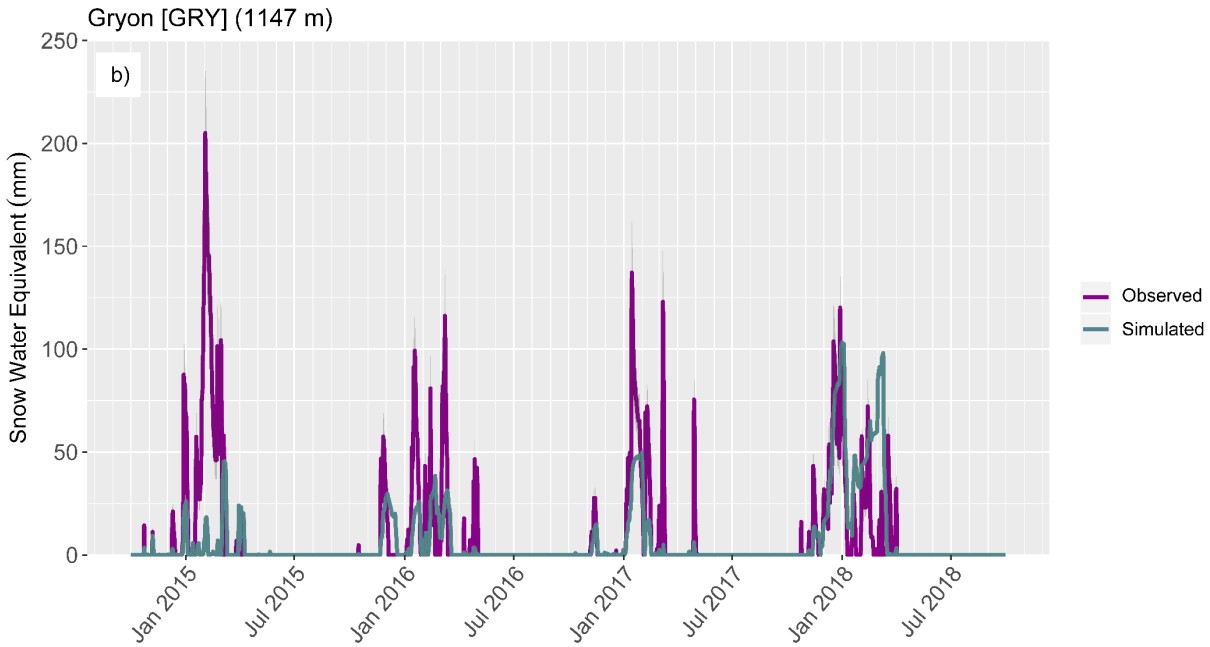

**Figure 5: Observed and simulated Snow Water Equivalent (SWE) time-series at the two measurement stations, Fully Grand Cor (a: COR, hourly) and Gryon (b: GRY, daily). It must be emphasised that the observations are not direct measurements of SWE, but rather are reconstructions based on snow depth and other measurements at COR, and purely snow depth at GRY. The grey bar shows the ± 20% region around reconstructed "observed" SWE values; this is added to account for the pixel-point nature of the comparisons.**




Firstly, it must be reiterated that the "observations" are in fact themselves modelled reconstructions, and that two different methods were used at the stations depending on data availability. Furthermore, the observed data correspond to discrete

station locations whilst the modelled values correspond to the 25 m pixel (average) within which each station was located. Therefore, should the terrain at the precise station locations not be perfectly reflective of its immediate surroundings, this represents one potential, entirely expected source of mismatch. In an attempt to represent this possibly, a region corresponding to ± 20 % around the observations is shown in Figure 5, this being approximately the maximum SWE mismatch that one could expect for this reason (T. Jonas, *personal communication*). In other words, if a given model is able

to reproduce the observations approximately within this range, then it could in fact be a perfect simulator. (In reality, of course, no model is perfect; all are associated with some inherent biases and imperfections). Overall, the dynamics of the snowpack evolution are replicated rather well, including the contrast between the lasting seasonal snowpack at the higher elevation station (COR) and the much more intermittent pattern at the lower elevation one (GRY). There does, however, appear to be a general tendency towards underestimation. Across both stations, the observations from the most recent winter,

2017/2018, are best reproduced.

## 5.2. Estimated parameter values

The parameter values estimated via inversion can be considered as "results" in their own right. Two in particular are interesting to briefly consider. Firstly, the high value of 1.45 taken by the wind speed-independent snow correction constant, *snob,* which actually reached the upper bound that was permitted in this study, attests to the considerable underestimation

bias generally contained within the winter station precipitation measurements. It must be remembered that in the approach applied herein, this constant factor is combined with the wind speed-dependent factor, *snoa*, which was estimated to be 0.0283 (i.e. an increase of 2.83 % per additional m s$^{-1}$ of wind speed), and the interpolated wind speed to determine effective precipitation. Secondly, *idwedr* took on its lowest permitted value, i.e. improved fits were obtained when a strong elevational gradient was enforced upon the interpolated precipitation fields.

## 5.3. Snowpack evolution

A key benefit of the model developed, and indeed any distributed, transient simulator, is that is "fills in the gaps" in space and time between the available spatial and temporal observations. In this spirit, Supplementary Video S1 presents an animation of the simulated evolution of SWE at daily time-step during Winter 2017/2018. The redistribution of snow from steep slopes is particularly apparent in the animation, and are the zones in which this redistribution is simulated are

consistent with our field experience during this period.

## 5.4. Hydrological plausibility: arrival of water at the land surface and potential evapotranspiration

Figure 6 illustrates the spatio-temporal distribution of both the arrival of water at the land surface (comprised of liquid precipitation, snowmelt, firn melt and ice melt) and $ET_P$ over the last two simulated hydrological years. As expected, very





little meltwater input is observed during the winter months, when temperatures are generally below freezing. Conversely, the highest meltwater volumes are produced during the spring melt, especially the months of April, May, and June. The increase in the elevation at which the majority of melt water is produced as the melt season progresses, and the localised contribution of the glaciers during the summer months, are also discernible. Liquid precipitation during the summer and autumn months, which can be highly concentrated in space and time in such environments, is averaged out in these plots (appearing as a fairly low and constant daily mean value in non-glacierised areas). For $ET_P$, a strong pattern according to seasonality and elevation is apparent – with low values everywhere in winter but restricted to high elevations in summer.







**Figure 6: Spatio-temporal patterns of (a) liquid water arriving at the surface (i.e. liquid precipitation + snowmelt + firn melt + ice melt), and (b) potential evapotranspiration ($ET_P$) generated using the optimised model configuration over the two hydrological years 2017-2018. In both cases, the underlying hourly data are expressed as daily mean values averaged across each calendar month, in metres (m).**



The maps in Figure 6 do not intend to imply that the water arriving at the land surface and $ET_P$ varies smoothly within these monthly periods; this is absolutely not the case. To illustrate this, in Supplementary Figure 5, the components of surface water input and potential evapotranspiration are presented at hourly time-step (averaged across the catchment) for the period January to October 2018 are presented. Pronounced diurnal cycles in $ET_P$ are evident, with both peak magnitudes and
amplitudes occurring during summer. The contrasting importance of the two components of "snowcover outflow" during different periods of the year is also apparent; i.e., the diurnal snowmelt cycles during the spring, and then the higher but more infrequent peaks in summer and autumn that are associated with rainfall events. Plotted on the same scale, the contributions from the glaciated areas (again averaged across the entire catchment) are barely discernible.

To further verify the hydrological plausibility of our results, simulated "snowcover outflow" and measured discharge were compared. More specifically, in Figure 7(a) hourly, spatially-averaged "snowcover outflow" is plotted alongside hourly normalised discharge from the Vallon de Nant sub-catchment (measured at the Avançon Weir; see Fig. 1) over spring 2018 (April to June inclusive). A straightforward comparison of the two cumulative totals produces an estimated runoff ratio of 0.61 within this three-month period, although this value is tentative given uncertainties associated with precipitation, the
model, and observed discharge. As expected, at the diurnal timescale, snowmelt leads increasing streamflow. On slightly longer timescales, dependence remains present. For example, the decrease in measured streamflow just before the start of May coincides with a marked reduction in simulated water inputs. Extending this plot to later in the summer (not shown) revealed that a certain proportion of the "excess" spring melt inputs arrive in the stream later (which can be considered the "buffering" capacity of relatively shallow groundwater storage), whilst other components will be lost to actual
evapotranspiration and perhaps also deeper groundwater storage and/or groundwater exportation across the topographic divides. Figure 7(b), meanwhile, shows the relationship between these data aggregated to a daily time-step (the lagged and strongly dampened streamflow response relative to the melt inputs complicates such comparisons using the hourly data). A power-law relationship appears present, as illustrated by the line estimated using non-linear least squares regression. Some hysteresis in the relationship, which would be expected as groundwater storage and subsurface saturation increases during
the melt season, can also be seen.



**Figure 7: a) Hourly catchment-averaged simulated "snowcover outflow" (i.e. snowmelt from non-glaciated areas plus any liquid precipitation) vs. hourly observed stream discharge for the Vallon de Nant sub-catchment for spring 2018 (discharge gauged at Avançon Weir and normalised according to catchment area), and (b) daily sum simulated snowcover outflow vs. daily mean**
**normalised observed discharge, again at the Avançon Weir station, for the same 3-month period.**

These lines of evidence therefore further reinforce the hydrological plausibly of our snow simulations, and therefore provide confidence that the datasets generated will form suitable inputs for one planned subsequent application; namely the development of a sophisticated, fully-integrated surface subsurface-hydrological model. To this end, they will be integrated with a 3D model of bedrock geology (Thornton et al., 2018) and recent geophysical surveys.





**5.5. Predictive uncertainty and data worth**

Table 4 shows the estimated pre- and post-calibration uncertainty standard deviation of the selected predictions. For all predictions, the uncertainty associated with the prediction in question is substantially reduced (by a factor of approximately four) by the calibration process.

| Prediction | Pre-calibration uncertainty standard deviation | Post-calibration uncertainty standard deviation |
|---|---|---|
| **SWE 01/04/2015** | 219.09 (mm) | 57.87 (mm) |
| **SWE 01/04/2016** | 200.81 (mm) | 57.82 (mm) |
| **SWE 01/04/2017** | 628.54 (mm) | 151.63 (mm) |
| **SWE 01/04/2018** | 320.29 (mm) | 82.13 (mm) |
| *$F_1$ 22/05/2017* | 0.0537 (-) | 0.0365 (-) |

**Table 4: Pre- and post-calibration uncertainty standard deviation of the selected predictions.**





**Figure 8: (a) Parameter contributions to predictive uncertainty variance pre- and post-calibration for the predictions of Snow Water Equivalent on 1 April 2016, and (b) snow pattern (summarised by $F_1$) on 22 May 2017.**

Figure 8 provides an indication of contribution of the different model parameters to the uncertainty variance, both before and after calibration, for two of the five selected individual predictions shown in Table 4. In these plots, the uncertainty variance contributions have been normalised with respect to the pre-calibration uncertainty variance associated with the respective predictions. Figure 8(a), which deals with the prediction of SWE on 1 April 2016, firstly reveals that there are numerous parameters that do not contribute (or do so only negligibly) to predictive uncertainty either before or after calibration. In

other words, the prediction is insensitive to these parameters. A slight reduction in the contribution to uncertainty variance can however be observed for the parameters *idwedr*, *snob*, and *snoa*. The results for the other three 1 April SWE predictions were similar, and so are not presented in the interests of space. For the prediction of the spatial snow extent on the 22 May 2017, summarised by the $F_1$ statistic (Figure 8(b)), practically all parameters make some discernible contribution both before and after calibration. Interestingly, a large reduction in the post-calibration uncertainty associated with *lwin* is again

observed, but the post-calibration uncertainty associated with *lwou* in relation to this prediction is actually higher that the





pre-calibration value. For all other parameters, the uncertainty contribution post-calibration is very similar to the pre-calibration level, suggesting a certain insensitivity of the simulated snow extents to varied parameter values.

Figure 9 provides two alternative representations of the worth of the observations belonging to the five different groups in
the calibration process. Figure 9(a) shows the increase in post-calibration predictive uncertainty variance, relative to pre-calibration uncertainty variance, associated with each of the 5 selected predictions incurred by omitting each observation group from the calibration dataset in turn. Removing either $F_1$, $F_2$, or $F_3$ is observed to have very little detrimental effect on any of the predictions. Figure 9(a) also reveals the notable contribution that both time-series, but most especially that at GRY, make to the prediction; the uncertainty variance of this prediction increases markedly if these time-series groups are
removed. Finally, Figure 9(b) provides an indication of the decrease in uncertainty variance relative to the pre-calibration uncertainty variance accrued when each observation group comprises the sole member of the calibration dataset.







**Figure 9. (a)** Increase in relative (to the pre-calibration uncertainty variance) post-calibration predictive uncertainty variance associated with each of the 5 selected predictions that is incurred by omitting each observation group from the calibration dataset in turn, and **(b)** decrease in relative (again to the pre-calibration uncertainty variance) uncertainty variance accrued when each observation group comprises the sole member of the calibration dataset. The "redundancy" or commonality of information between the three spatial observation groups (i.e. $F_1$, $F_2$, and $F_3$) is clearly apparent.

Once again, including any one of the observation groups $F_1$, $F_2$, or $F_3$ (alone) in the calibration datasets leads to only modest reductions in the uncertainty variance associated with any of the predictions (although this is not to say that in combination they do not have a more pronounced effect). Inclusion of only either the COR or GRY observations in the calibration dataset leads to similarly large reductions in the uncertainty associated with the prediction of SWE at COR. The uncertainty around the $F_1$ prediction on 22 May 2017 is also greatly reduced by including either of the groups, although only by about half as much as the reduction seen for the 1 April SWE predictions.





# 6. Discussion

## 6.1. Comparison with F-statistics reported in previous studies

Figure 10 compares the F-statistics obtained in the present study to the small number of equivalent statistics (i.e. based on Landsat images and identical metrics) that have previously been reported. The data that underlies these plots are presented in Supplementary Table 2. More specifically, three previous known studies have presented F-statistics to quantify the fit between a distributed snow model and Landsat imagery. However, all did so for only a small number of days (between one and three); Schöber et al. (2010) simply presented $F_1$ values corresponding to two days, but did so for numerous different

catchments, while Bernhardt et al. (2012) and Warscher et al. (2013) provided all three F-statistics for their respective study catchments and selected days. In contrast to the present study which also included mid-winter and late summer days in the catalogue of calibration images, the previously published F-statistics all correspond exclusively to the spring and early summer periods when the catchments in question are only partially snow covered. As such, to enable the fairest comparisons possible, only our F-statistics from our studies that also correspond to spring and early summer were considered.

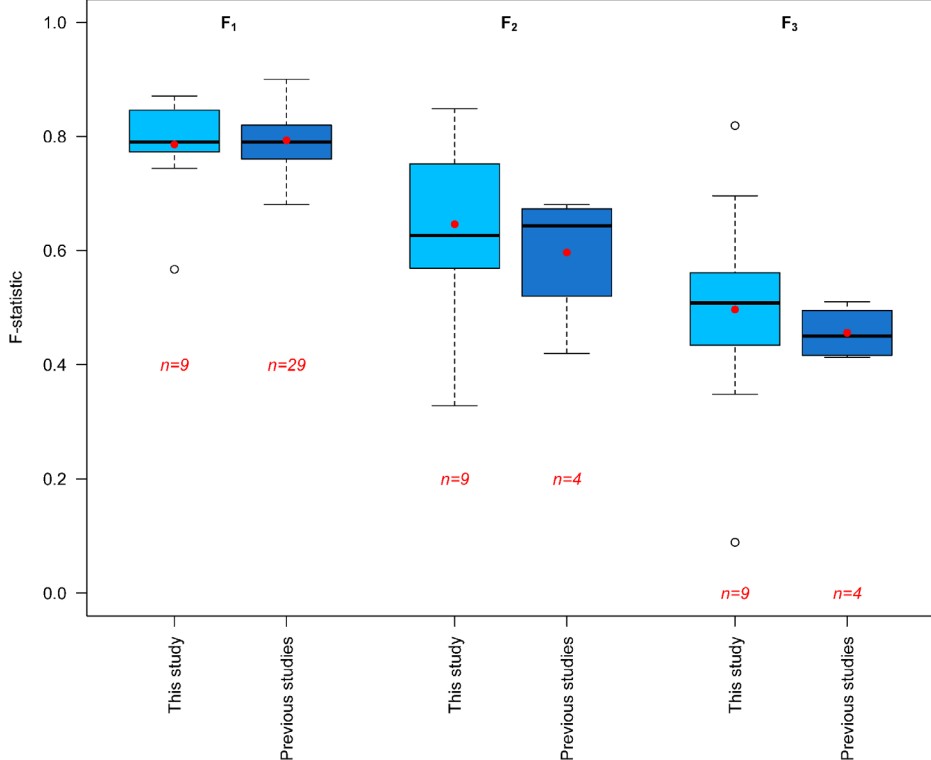

**Figure 10. Comparison between the spatial model fits, quantified using F-statistics, obtained in the present study and those reported by all previous known studies that applied the same metric on the basis of Landsat imagery. The medians are shown using a thick black line in the traditional fashion, whilst the means are also indicated by the red dots. For the fairest possible comparison, only the F-statistics from the present study that corresponded to the spring and early summer period (i.e. 08/04/2015,**
**24/04/2015, 10/04/2016, 13/04/2017, 20/04/2017, 29/04/2017, 22/05/2017, 25/05/2018, 19/06/2018) were included in this plot. The data that underlies these plots are presented in Supplementary Table 2.**



Although the number of observations in certain classes is rather small, some interesting observations can made. Firstly, in the case of all statistics, the range of our F-statistics broadly overlap with, and perhaps even have a tendency to be higher than, those of pervious values, even if the $F_1$ mean and median are marginally lower than those calculated from the previously published values. This tendency towards higher values generated in this study appears most pronounced in the cases of $F_2$ and $F_3$, and in both cases our mean values from the present study are higher than their published counterparts. Interestingly, with the possible exception of the study of Schöber et al. (2010) who may have used the spatial Landsat snow observations in an informal fashion to adjust certain parameters in their snow model (ambiguity remains because this step was not fully explained), the previous studies only used this spatial data in model evaluation. This is of course in contrast to the present study in which the spatial data form calibration targets.

With this in mind, although it is slightly disappointing that the explicit calibration did not yield higher $F_1$ scores, it may be argued that a real benefit of calibration can be seen in the noticeably higher $F_2$ and $F_3$ scores. Indeed, it may be recalled that these metrics were assigned enhanced weight in our calibration processes. Additionally, our calibration process was not solely informed by the maps, but rather sought to seek an acceptable balance between fits according to both the maps and the SWE time-series. It is likely that better spatial fits could have been achieved if the maps alone formed the calibration target, but probably with an adverse effect of the accuracy of simulated snow water storage across the catchment. In summary, although a fuller catalogue of previously published F-statistics would be required in order to assess the statistical significance of these results, they demonstrate the general appropriateness – and perhaps even added value – of our approach. Lastly, given the variably in our F-statistics between days, it is unclear how consistent in time the generally good performance of the previously published models might be (sine they only report statistics from a small number of days). A useful feature of our study is therefore the larger number of days for which the statistics were compiled.

### 6.2. The magnitude of undercatch correction

The magnitude of the solid precipitation correction factor estimated in this study is broadly consistent with existing literature. For instance, Sevruk (1985) suggested that overall, precipitation in Switzerland is underestimated by between 4% (in summer at low elevations) and approximately 40 % (at high altitudes in winter). The study of Pan et al. (2016) found the sheltering effect of surrounding vegetation to be an important influence on the magnitude of any underestimation in northern Canada, with well-sheltered sites requiring much less correction. At open sites, the bias corrections applied increased annual precipitation by between 15 and 34 %. In windy sites with a high proportion of snow, even greater corrections can be necessary – even exceeding 50% (*Ibid.*). Finally, in their simulation of a 200 km$^2$ region of Switzerland, Bavera et al. (2014) applied a fixed 30 % factor to solid precipitation.





### 6.3. Potential sources of residual mismatch

Uncertainty in the observed data aside, a large proportion of the remaining spatio-temporal mismatch between our simulations and observations is likely to be associated with the meteorological forcing data. In light of the combination of
relatively low station density and variable but sometimes high frequency of data gaps (Supplementary Figure 2), the interpolated spatial fields of meteorological variables are undoubtedly uncertain (and potentially even erroneous). More specifically, it must be emphasised that whilst the temporal data coverage and therefore crossover of the meteorological data varies throughout the simulation period, the parameters in the model are of a global nature. Accordingly, the rain-snow threshold temperature (*rstt*) and solid precipitation correction factors (*snoa* and *snob*), for example, are applied constantly in
time and space through the simulation domain. It follows that when these parameter are estimated though the calibration processes, best overall values with respect to the data are produced. In reality, however, the error distribution associated with precipitation measurements actually varies on a station-by-station, event-by-event basis. The spatial dependencies in these parameters away from the station locations, which the interpolation processes attempts to recreate, also probably demonstrate spatial non-stationarity. In other words, it may be that the model structure, by not allowing any bespoke
correction on a per-station/event basis, is insufficiently flexible to fully compensate for deficiencies in the meteorological measurements during certain periods/at certain locations. In this sense, perhaps improved fits could have been achieved by simply scaling relatively complete time-series measured at (an) individual location(s) using linear, elevation-dependent relationships, although this approach would be less satisfactorily during periods with high meteorological data availability since much of it would essentially have been discarded.

### 6.4. Predictive uncertainty and data worth

The presence of a large number of parameters that do not contribute to either the pre-or post-calibration uncertainty in the SWE prediction (Figure 8) is unsurprising since most of these parameters concern the gravitational redistribution component of the model, whereas the COR measurement station will have been strategically sited so that the measurements are very rarely affected by such processes. Another striking feature of the plot is the large reduction in the predictive uncertainty
associated with the longwave correction parameters, *lwin* and *lwout*, induced by calibration. The rather counter intuitive situation whereby the post-calibration contribution of a particular parameter to predictive uncertainty actually exceeds its pre-calibration level can occasionally arise when a parameter to which the prediction is insensitive can only be made in conjunction with another parameter to which the prediction is indeed sensitive; see Docherty (2010) for further explanation. Additionally, the "robustness" indicated by the similarity between pre-and post-calibration parameter contributions to
uncertainty to the spatial prediction (Figure 8(b)) could be particularly beneficial in applications where the spatial snow patterns, rather than volumes of water stored, constitute the key information (as perhaps in vegetation species distribution predictions).



Turning our attention to Figure 9(a), the fact that removing either of the "F" groups form the calibration dataset has little

adverse effect on any of the predictions can be explained by the fact that when one of the above groups in omitted, very similar information remains in the other two groups. The comparatively small number of observations in these groups coupled with the uniform weighing applied to all observations could also partially explain these results. It is also interesting to highlight that four of the five predictions under consideration correspond to the SWE predicted at the high elevation COR. In light of this, the analysis suggests removing the data at GRY from the calibration dataset has a more detrimental effect

than removing the other observations at COR (the very location of the prediction). The apparent significance of the GRY data is especially notable given that the number of observations at this site is substantially lower than at GRY (due to lower measurement frequency). It is likely that being straddled more frequently by the 0°C isotherm, the SWE time-series at GRY contains more important information about temperature (and therefore temperature gradients) and snow limits than the COR data, in which distinct accumulation and ablation seasons are apparent. The notable contribution of both time-series, but

especially that at GRY, apparent in the same figure shows the importance of obtaining alternative types of data and employing them within a multi-objective approach. This result is consistent with the conclusion of Tuo et al. (2018), who showed that SWE data can also be included to good effect in the calibration of hydrological models in alpine catchments. The result presented in Figure 9(b) – that even including only one of the time-series as the sole calibration dataset substantially reduces the uncertainty in the spatial prediction – is indicative of an important "flow of information" from the

time-series to the predicted spatial snow patterns. This result can probably be generalised to the other days on which spatial simulated spatial patterns were compared with observations.

Finally on the subject of uncertainty, since an important overriding aim of the present study was to generate the best possible inputs for subsequent hydrological modelling that coincide in time with other measurements from the study region (e.g.

groundwater levels; not shown), no snow observations were specifically withheld for evaluation. In some senses, this represents a limitation of the present work. Future research should certainly explore the influence of the calibration specific period, and/or assess model performance under different conditions. That said, the uncertainty analysis is an alternative to a more traditional split-sample model evaluation.

## 6.5. Wind redistribution

Spatially distributed wind drift correction factors have been successfully applied in an attempt to account for the influence of wind transport processes on snowpack heterogeneity (e.g. Hanzer et al., 2016; Marshall et al., 2019). However, as mentioned earlier, after extensive testing, the wind redistribution algorithm avaliable in WaSiM (developed by Warscher et al. (2013)) was not applied in our final model; doing so was found to substantially reduce model fits with respect to the observations. The algorithm computes a temporally invariant, spatially-distributed grid of correction factors that are then applied to the

interpolated precipitation fields such that precipitation falling on predominantly sheltered slopes and on the leeward side of ridges is augmented, whilst that falling on exposed slopes is reduced. In order to generate such a grid, a single prevailing




wind direction (more specifically, sector) must be specified. An analysis of the relationship between the speed and direction of winds in winter at high elevations stations within our broad study area was conducted. The criteria of "winter" and "high elevation" were used to subset the data since it is under these conditions that snow redistribution by wind is of most concern. The results, presented in Supplementary Figure 6, revealed that no such single strong winter wind direction prevails. Rather, strong winds can apparently can originate from contrasting directions, likely a function of larger-scale synoptic meteorology. Some influence of the local topography can also be seen. The range of the calculated wind redistribution factors also seemed somewhat high (leading to both too much "erosion" and too much "deposition"). In addition, and in contrast to the gravitational redistribution algorithm, the wind redistribution approach does not conserve mass within a given area (e.g. a catchment). In summary, it may simply be that in mountainous regions where large scale meteorological systems combine with extremely complex topography to produce considerable spatio-temporal variability in wind fields, relatively simple algorithms of this nature cannot be expected to match site specific, highly resolved, and information-rich snow observations. As such, the development of extended empirical approaches that perhaps account for measured high-elevation wind direction and/or are calibrated explicitly to observed redistribution magnitudes in the vicinity of ridges could form an appropriate intermediate objective, at least until snow transport can be simulated physically at catchment scale. In any case, in our study site at least, we are confident that snow redistribution by wind is of secondary hydrological importance to gravitational redistribution.

**6.6. Ongoing debates regarding model calibration**

Somewhat more generally, there remains some debate in the literature concerning the most appropriate approach for including snow in the calibration or evaluation of hydrological models. On the one hand, it has been argued that since the volumetric information contained within discharge measurements is complementary to the internal spatial pattern information present in distributed snow observations (Finger et al, 2011), these two types of observations should be considered simultaneously. This approach was pursued by, for example, Finger et al. (2011), Sreathsea et al. (2014), and Deuthmann et al. (2014). The argument runs that the constraint provided by discharge can help ensure that the aggregate simulated amount of water in the system is approximately correct, which is important given the uncertainties associated with precipitation measurements and products, whilst observed snow pattern constraints help ensure that this runoff is being generated in the right areas. On the contrary, it has been posited that calibrating such models is better tackled in a linear, sequential fashion whereby certain components – in this case the snow simulations – are initially dealt with. Only then does one proceed to simulate discharge and any other hydrological variables of interest. The principal argument in favour of this approach is that simultaneous calibration may allow for error compensation (Ragettli and Pelliccciotti, 2012; Magnusson et al., 2015) which may be hidden or easily missed in the modelling processes (at least unless extremely careful internal evaluative work is conducted).



Here, the view was taken that if a given set of snow observations are sufficiently informative, then using them alone as a constraint in what may eventually amount to the first step in a wider, sequential calibration processes should be enough to ensure that the volumes of water are reasonably accurate. Our hydrological plausibly assessment certainly suggests that this is the case. Indeed, for this very reason both a broad range of spatial and volumetric (SWE) time-series information at highly contrasting sites were utilised in the snow model calibration. In this way, the chance of parameters related to the surface or subsurface properties in any subsequent compensating for poor snow simulations is minimised. Either way, neglecting to verify internal models states should be entirely avoided. An additional reason why the two aspects were separated here and "only" the former presented it that we intend to proceed to develop a comprehensive, integrated hydrological model to simulate the remainder of the hydrological cycles. Being somewhat more complex than standard approaches, this will require sufficient explanation and so falls beyond the present scope.

## 7. Conclusions

This paper has presented a novel approach to the calibration of distributed snow models in rugged alpine catchments. The physically-based core of the model enables pronounced spatio-temporal variably in energy balance components, which is largely responsible for heterogeneous snow patterns and therefore melt rates in such terrain, to be explicitly captured. Physically-oriented snow models are moreover likely to generate more reliable predictions under modified climatic conditions than their simpler counterparts. That said, processes like gravitation snow redistribution can also substantially influent meltwater patterns in steep regions, yet currently cannot be represented on an entirely physical basis at catchment and larger scales. A pragmatic empirical gravitational redistribution algorithm therefore was applied. Substantial additional uncertainties related to biased precipitation measurements, for instance, necessitated the introduction of further parameters.

Reliably reconstructing the evolution of alpine SWE therefore hinged on the non-trivial task of parameter estimation. The challenge was addressed via the development of a multi-objective calibration approach that incorporated complementary, high-resolution snow observations. A component of the objective function was explicitly spatial in that misfits at the pixel level were penalised. Indeed, to our best knowledge, this study represents the first time a high-resolution, distributed snow model has been calibrated in this fashion using snow observations derived from Landsat data. An initial conclusion that may be drawn is that the standard threshold of 0.4 proposed in the literature for the generation of binary observed snow maps using NDSI calculated from satellite imagery should not be applied indiscriminately; much lower thresholds that were variable between images were required here to produce reasonable snow extents.

As a result of our approach, snow dynamics could ultimately be satisfactorily reproduced, with the spatial fit metrics obtained comparing favourably with the few presented previously. However, substantial corrections to measured winter precipitation totals to account for wind-induced undercatch were necessary to achieve this. Aside from observational



uncertainties, much of the remaining model-data mismatch is likely to be associated with the meteorological forcing data – in terms of the measurements themselves and the spatial interpolation process. As such, efforts to better understand and account for the uncertainties and biases inherent with mountain meteorological measurements and gridded data products should continue to form a research priority. A subsequent uncertainty and data worth analyses indicated that i) the

uncertainty variance of indicative predictions of snow states, both spatial and volumetric, were substantially reduced through calibration, ii) calibrating two parameters that are applied in the model adjust the longwave component of the surface energy balance, and thus compensate of errors in cloudiness and albedo where necessary, was especially beneficial, and iii) SWE time-series at the lower elevation station was particularly informative in the calibration process despite the comparatively small number of observations at this site.


More generally, this study has shown that snow maps developed from Landsat 8 data can be incorporated into a spatially explicit objective calibration framework to constrain simulations of snow dynamics and ultimately generate hydrologically plausible meltwater datasets in rugged terrain. Furthermore, a generic (i.e. model and concept-independent) approach to quantifying the uncertainties and data worth associated with snow model predictions – a rarely attempted task in snow

hydrology – that is both feasible and useful has been demonstrated. Such work should be undertaken more routinely as part of snow simulation exercises. Finally, datasets such as those generated herein hold considerable potential to inform the next generation of distributed hydrological modelling efforts in alpine terrain; work in this direction is urgently required to improve our holistic understanding of alpine hydrological system responses to ongoing climatic change.

**Data availability**

All model-rated datasets and code that the authors are free to share have been uploaded to the following online repository: https://doi.org/10.6084/m9.figshare.9016154.v1 (Thornton et al., 2019). High-resolution versions of all supplementary figures are also available at this location, as is the supplementary video. The meteorological data obtained from MeteoSwiss, and the terrain and land cover data obtained from swisstopo remain subject to restrictions. Therefore, the processed model inputs relating to meteorology, terrain, and land cover are not included in the folder. Requests for these datasets should be

directed to the respective organisation. Third-party executables are similarly excluded, but are obtainable as follows: WaSiM executables and documentation can be downloaded from http://www.wasim.ch/en/products.html, whilst PEST executables and documentation can be downloaded from http://www.pesthomepage.org/Downloads.php

**Author contributions**

J.M.T. conducted the vast majority of the work, including obtaining and processing the meteorological, satellite, and GRY

SWE time-series data, establishing the model, designing and implementing the calibration approach, preparing the figures,



and drafting the manuscript. T. B. conducted the simulation using the SNOWPACK model at the COR station and provided guidance regarding some of the meteorological data. G.M. and P.B. provided advice at all stages. All authors participated in the redaction of the manuscript.

**Competing interests**

The authors declare that they have no conflict of interest.

**Acknowledgments**

This work was conducted as part of the IntegrAlp project, funded by the Swiss National Science Foundation (SNF project CR23I2_162754). The authors thank B. Schaefli for useful discussions and J. Schulla for answers to questions regarding the use of WaSiM. Funding for the weir was provided by the University of Lausanne, ETH Zürich, and the Swiss Federal

Institute for Forest, Snow and Landscape Research (WSL). The authors declare no competing financial interests.

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
