# Peer review of "Efficient multi-objective calibration and uncertainty analysis of distributed snow simulations in rugged alpine terrain"

_The Cryosphere, 2019_

## Referee Comment (RC1) · Anonymous Referee #1 · 26 Oct 2019

This paper presents an uncertainty analysis of snow simulations with the model WaSiM in a mountainous region in western Switzerland using multi-objective pre- and post-calibration methods including the usage of Landsat 8 images. The paper is well-written and the authors took especially great care in producing good und understandable figures as well as supplementary material. Moreover, it seems that the authors have a comprehensive overview on literature close to their study. However, on the one side as mentioned by the authors themselves, the absence of 'real' continuous snow measurements within the study site is a clear limitation of this study, but on the other side the authors try the best in including 'reconstructed snow measurements'. I have some general and some specific points, which should be considered carefully before publi-

cation.

General: - The results should also be separated in snow accumulation and snow ablation phase (as you shortly mentioned the two phases on p.38, l.819 and in the abstract). I guess there would be then distinct differences in your calibration. This should be presented and discussed.

- In my opinion, the results and discussion part are not a clearly separated. In both sections, points of results and discussion can be found. Maybe you merge these two sections or make a clearer separation into results and discussion.

- It comes far too late in the manuscript that you are using the model WaSiM. This should already be mentioned in the abstract and the introduction.

- Snowpack simulations cannot be described as 'actual' measurements!

- Just a very general remark: Why don't you use additionally Alpine3D and compare it to your results?

- The paper is very extensive and long. Some passage are only descriptions and repetitions of other literature. I would suggest to shorten (in total up to 3 pages) several points especially in the sections introduction, data and methods and to focus more on the relevant points of your work.

- It is irrelevant which type of processing software (e.g. R) you used. . .

Specific:

- p.6, l.147: addressed with 'd'

- p.6, l.148: stations without 's'

- p.6, Please insert which snow model you used. I guess you are talking about WaSiM? But can this really be described as a snow model?

- p.8, Figure 1: You mentioned the two headwater sheds Vallon de Nant and Vallon de

La Vare – please mark them in Figure 1.

- p.9, Section 3: In the section 'Data' you also describe the processing / generation of the data. Please consider renaming this section accordingly.

- p.9, Section 3.1: Start by describing which input the model needs. The information is just somewhere in the following text.

-p.10, Eq.1: please insert a reference for Eq. 1

- Section 3.2: Here information on the applied snowpack model (I guess SNOWPACK (Lehning et al. 2002) is missing. Which were the input data for the SNOWPACK simulations? Actually, you need specific meteorological input data (and not necessarily snow measurements). Why couldn't you use the SNOWPACK simulations at the meteorological stations within your catchment? Please also think about using Alpine 3D (Lehning et al. 2006). And, if you used snow measurement data from instruments at the higher station, please specify, which snow sensors where used.

- p.11, l.261: '…gradients (with 's')

- p.13, Figure 2: In my opinion this figure is not needed. Anyway, in Fig. 2b, the classes were not rounded to e.g. 2 digests and the classes seem to be a bit random; in Fig. 2c, the legend does not correspond to the image (no snow should be white) and to the colours in Fig. 3.

- p.11, l. 282: Why did you exclude all cloud covered Landsat 8 data. Maybe there were some useful images, which were only partly cloud-covered. It should be discussed why you prefer using only cloud-free data whilst allowing a lesser spatial resolution in potential available images.

- p.15, l.338f: IDAWEB? IMIS? Not clear to all readers please describe acronyms.

- p.15, l.345ff: Which parameters for the station was used by applying Jonas et al (2009)?

- p.15, l.354: I strongly disagree that these are actually direct snow measurements. This has to be corrected.

- Section 3.3: What is the temporal resolution of the streamflow data?

- p.16, l.362: Define 'rating curve construction'.

- Section 4: As you actually describe some methods also in Section 3, I would suggest to rename this chapter and relate it especially to your modelling, calibration and uncertainty estimation 'work' with WaSiM.

- p.16/l.365: without 'to'

- p.19, Table 2: This table is not necessary and can be replaced by one sentence in the text. - p.20, l.486ff: Is there a reference for the chosen weighting? Or why did you choose the weighting of 60:40?

- p.22, l.537: 'were also developed': I suggest to write 'were also calculated'

- Section 5: Please rethink your subchapter captions and a general merge with Section 6.

- p.26, l.603ff: For example, this point belong more to the discussion...

- p.30, Figure 7a: Please explain more clearly what is compared and shown in this graph.
* * *

---

## Author Comment (AC1) · 11 Nov 2019

*Reviewer comments are in blue, and responses are in **black**.*

Firstly, we would like to thank Anonymous Reviewer 1 for the positive and constructive comments.

We provide this initial response to open the discussion and outline the changes we plan to make when we revise the manuscript. We will wait for the additional review(s) to become available before actually implementing these changes, however.

**This paper presents an uncertainty analysis of snow simulations with the model WaSiM in a mountainous region in western Switzerland using multi-objective pre- and post-calibration methods including the usage of Landsat 8 images. The paper is well-written and the authors took especially great care in producing good und understandable figures as well as supplementary material. Moreover, it seems that the authors have a comprehensive overview on literature close to their study**.

We are grateful that the efforts made to present a well-written paper with sufficient background context, appropriate figures, and extensive supplementary material were appreciated by the reviewer.

**However, on the one side as mentioned by the authors themselves, the absence of 'real' continuous snow measurements within the study site is a clear limitation of this study, but on the other side the authors try the best in including 'reconstructed snow measurements'**.

The concern about 'real' continuous snow measurements may be a semantic issue. To be clear, we used continuous measurements of snow depth at two stations which were located in the vicinity of the target catchments. The simulation domain was extended in order to include these locations. In our view, these measurements can be described as 'real' continuous snow measurements.

To enable more meaningful comparisons to be made between our simulations and the available (time-series) observations, measured snow depths were converted into Snow Water Equivalent (SWE) via a modelling process. This process was different at each station according to data availability. Other studies using WaSiM did not attempt this extra step, and simply presented comparisons between simulated SWE and measured snow depth (e.g. Warscher et al., 2013).

In fact, some continuous snow depth measurements were attempted at the internal stations, but generally yielded gap filled and quite uncertain data that were not considered suitable for inclusion in our calibration process. That said, we certainly agree that more extensive in situ measurements – especially repeated SWE surveys at different locations in the study area – would have been extremely valuable. Unfortunately, the steep and remote (no ski installations etc.) nature of the study catchments mean that avalanche risk can considerable yet uncertain, and so fieldwork in winter and spring strongly limited. Faced with these challenges, as the reviewer highlights, we tried our best to develop datasets that would be as informative as possible in constraining our model.

Incidentally, a possibility for future work would be to undertake LiDAR scanning under summer and then winter conditions (e.g. Cochand et al., 2019) in order to generate (a) high resolution snapshot(s) of snow depths. Of course, a helicopter flight would be required to generate a catchment scale map, which brings cost implications, and furthermore a model to predict density would still be required to convert to water equivalent.

**I have some general and some specific points, which should be considered carefully before publication.**

**General:**

**The results should also be separated in snow accumulation and snow ablation phase (as you shortly mentioned the two phases on p.38, l.819 and in the abstract). I guess there would be then distinct differences in your calibration. This should be presented and discussed.**

As mentioned in the abstract, it is true that both snow accumulation and ablation need to be accurately simulated if reliable patterns of meltwater arrival at the land surface are ultimately to be generated. Clearly, some of the parameters subjected to calibration relate only to one of these two phases. For instance, the undercatch correction factors purely affect accumulation, whilst the longwave correction parameter affects ablation. However, other parameters are shared between these process components (e.g. rain snow threshold temperature, which also dictates whether melt can occur if the energy balance is favourable) or neither of them (e.g. the parameter concerned with the redistribution module). Additionally, the moderate elevation of the catchment meaning that parts of it can be affected be repeated cycles of accumulation and melt within a single winter season. This meant that even dividing the simulated period up into distinct accumulation and melt periods would not have been straightforward. As a result of these two facts, we elected not to separately calibrate the model for the accumulation and ablation phases, but rather sought parameters that gave the best overall representation of the dynamics according to the constructed objective function.

A related issue that we could have dealt with in a more sophisticated fashion is that fact that the catchment usually becomes snow covered very quickly in winter, whereas the melt in summer leads to a much more gradual evolution of snow extents. Therefore, during the onset of winter conditions, small errors in simulated snow onset (either slightly too early or too later, perhaps due to an insufficient air temperature measurements during a given period and attendant uncertainties in the interpolated grids) would lead to very large errors on these spatial statistics, but these would most probably have only relatively limited hydrological importance (due to the lower snowpack water storage early in the season). In contrast, any large errors in the simulated snow extent in spring would likely have much more significant hydrological implications. In theory, this issue could have been partially addressed by weighting the observed snow extent maps that correspond to the "end" of the season more highly than those corresponding to the beginning, although this was not done in the present work. If space permits, this may be highlighted in a revised version (but we are mindful of the recommendation of shorten the manuscript).

The above being said, we will still try and discuss the results of the model (in Section 5.1 of the original manuscript) more distinctly in terms of accumulation and ablation in the revised version.

**In my opinion, the results and discussion part are not a clearly separated. In both sections, points of results and discussion can be found. Maybe you merge these two sections or make a clearer separation into results and discussion.**

Thank you for this feedback. We propose to combine the Results and Discussion sections to overcome this problem and avoid repetition.

**It comes far too late in the manuscript that you are using the model WaSiM. This should already be mentioned in the abstract and the introduction.**

We agree, and shall alter the manuscript to state clearly in both the abstract and the latter part of the introduction section that the model was set up using WaSiM.

**Snowpack simulations cannot be described as 'actual' measurements!**

This was not our intention at all. Line 355 of the original manuscript actually stated that *"**neither** of the "observed" SWE time-series, which are presented in Figure 5 (alongside their simulated counterparts from the final model), are actually direct measurements."* So we are in agreement!

**Just a very general remark: Why don't you use additionally Alpine3D and compare it to your results?**

Alpine3D was investigated and tested during the early stages of this project. Following that, we identified that Alpine3D does not enable the estimation of glacial dynamics or gravitational snow redistribution, factors that were believed to be important for our study and following work. It was therefore not chosen as the principal code. Whilst it would certainly have been interesting to additionally use Alpine3D and compare the snow results with those generated by WaSiM, this unfortunately this lay beyond the scope of our study. As such, it represents an idea for future work, i.e. to explore in more detail whether more snow models with complexity but fewer processes (Alpine3D) or lower complexity but more processes (energy balance + redistribution in WaSiM) is to be preferred for simulations at catchment scale in steep alpine terrain.

**The paper is very extensive and long. Some passage are only descriptions and repetitions of other literature. I would suggest to shorten (in total up to 3 pages) several points especially in the sections introduction, data and methods and to focus more on the relevant points of your work.**

We will attempt to shorten the paper as much as possible whilst retaining information that we consider crucial. The removal of Figure 2 and Table 2, as proposed by the reviewer (see below) will also help in regard to length.

**It is irrelevant which type of processing software (e.g. R) you used.**

In certain instances, we feel that it can be helpful to highlight the software or packages with which certain analyses were conducted (e.g. to guide future work and to acknowledge the developers where software has been made open source). Nevertheless, in this case, we will remove all references to R (e.g. on lines 215 and 517) in the revised version of the manuscript. In any case, the software used will become apparent to readers who consult the supplementary material.

**Specific:**

**p.6, l.147: addressed with 'd'**

**p.6, l.148: stations without 's'**

Thank you for spotting these typographical errors. They will be corrected in the revised version.

**p.6, Please insert which snow model you used. I guess you are talking about WaSiM? But can this really be described as a snow model?**

The reference to WaSiM has now been added in this section (in response to an earlier point). The revised text at line 158 will read: *"Initially, a fully-distributed energy balance-based snow model that includes gravitational redistribution is established using WaSiM (Schulla, 2017) at high spatio-temporal resolution".*

In our opinion, WaSiM is a hydrological model with a strong snow module or component. In this study, "only" the capabilities of WaSiM for interpolation meteorological data, simulating snow and ice dynamics, and estimating potential evaporation were employed. In other words, we did not proceed to simulate the remaining hydrological processes culminating in stream discharge. Essentially, since we were using these components of WaSiM as one would use a more conventional snow model, we feel this description is appropriate.

**p.8, Figure 1: You mentioned the two headwater sheds Vallon de Nant and Vallon de La Vare – please mark them in Figure 1.**

The extent of the Vallon de Nant sub-catchment is already marked by a polygon on Figure 1. The extent of the Vallon de La Vare is not discussed further and so actually is not really important to the comprehension of the paper. For this reason we propose simply modifying the text to indicate that the Vallon de La Vare lies immediately to the north east of the Vallon de Nant.

**p.9, Section 3: In the section 'Data' you also describe the processing / generation of the data. Please consider renaming this section accordingly.**

We will rename this section *"Data availability and processing"* to reflect the fact that this section deals with both aspects, i.e. it outlines the data that was available and/or selected for use, and also describes the processing of that data for the purposes of the study.

**p.9, Section 3.1: Start by describing which input the model needs. The information is just somewhere in the following text.**

Thank you for this suggestion. In the revised version, this section will start with a sentence listing the meteorological data the model needs, such as *"The model requires gridded estimates of incoming shortwave radiation, precipitation, relative humidity sunshine duration, air temperature, vapour pressure, and wind speed."*

**p.10, Eq.1: please insert a reference for Eq. 1**

This equation comes from Schulla (2017). The reference will be added in the revised version (Line 229).

**Section 3.2: Here information on the applied snowpack model (I guess SNOWPACK (Lehning et al. 2002) is missing. Which were the input data for the SNOWPACK simulations? Actually, you need specific meteorological input data (and not necessarily snow measurements). Why couldn't you use the SNOWPACK simulations at the meteorological stations within your catchment? Please also think about using Alpine 3D (Lehning et al. 2006). And, if you used snow measurement data from instruments at the higher station, please specify, which snow sensors where used.**

The key input data for the 1D SNOWPACK simulation, which was undertaken at the upper IMIS station (COR), consisted of the following hourly measurements:

- air temperature
- relative humidity
- wind speed
- reflected short wave radiation
- surface temperature
- snow height

A key point to emphasise is that although the model can be run using total precipitation, in which case snow height is not required, directly measuring total precipitation in such terrain is fraught with difficulty. Indeed, for this reason, the COR station only measures *liquid* precipitation directly. The measured depth therefore becomes an important input in the modelled estimates of SWE.

The list of sensors deployed at the IMIS stations is not published to our knowledge. Therefore, following receipt of this review, we requested and received the information from the SLF:

**Snowheight**: Campbell SR50A
**Snow Surface Temperature:** IR AlpuG (customized infrared sensor, http://www.alpug.ch/pdf/pdf20_IR%20Beilage2000%20d.pdf)
**Snow and Ground Temperature:** Campbell T107
**Wind:** Young 05103 (customized version for high alpine applications)

**Relative Humidity:** Rotronic Hydroclip
**Air Temperature:** Campbell T107 (thermal isolated mounting in radiation shield, not ventilated)
**Reflected Short Wave Radiation:** Campbell CS300
**Liquid Precipitation:** Campbell ARG100

Given the impact that inserting this list would have on the length of the paper (with in our view only a minor enhancement to the comprehensiveness), we propose will ask the journal before deciding whether or not to do so.

SNOWPACK could not be run at the stations within our catchment because not all of the required parameters were measured at these stations. In addition, a physically based model as SNOWPACK is very sensitive to data quality and does not tolerate gaps (e.g. in the snow depth time series). Interpolating the missing input was therefore not considered as an option.

Please see our earlier response concerning why Alpine3D results were not presented.

**p.11, l.261: gradients (with 's')**

Thank you, this change will be made.

**p.13, Figure 2: In my opinion this figure is not needed. Anyway, in Fig. 2b, the classes were not rounded to e.g. 2 digests and the classes seem to be a bit random; in Fig. 2c, the legend does not correspond to the image (no snow should be white) and to the colours in Fig. 3.**

This figure was included in the original version to simply illustrate the process of developing a binary snow extent map from a Landsat image. We agree that this information is fairly standard. We therefore plan to remove this figure in the revised version of the paper. Importantly, we are happy to do this because the same information (i.e. true colour composite and delineated snow extent) will still be available to readers in Supplementary Figure 4 (for all 17 images). The length of the paper (another point raised) will also benefit from this change.

**p.11, l. 282: Why did you exclude all cloud covered Landsat 8 data. Maybe there were some useful images, which were only partly cloud-covered. It should be discussed why you prefer using only cloud-free data whilst allowing a lesser spatial resolution in potential available images.**

The Landsat images used were not entirely cloud free, but crucially were cloud free over our study catchments. The manuscript will be updated to highlight this (line 285). We felt that this was important because our study area is relatively small in comparison with the scale of typical cloud features. In other words, if clouds were present over some of our area, then in percentage terms this would likely be considerable.

Our scripts quantifying the match between observations and simulations and weighting scheme employed in the calibration would then have had to have been substantially altered to account for the variable image coverage. Especially given the careful procedure that was followed to define bespoke NDSI thresholds for each image, we feel that 17 images in total, spanning the full range of snow cover conditions, were sufficient to demonstrate the concept of the "granular" calibration metrics and evaluate the ability of the model to simulate varied snow cover conditions.

A final note is that excluding cloud affected images as we did of course results in a lower *temporal* (not spatial) resolution.

**p.15, l.338f: IDAWEB? IMIS? Not clear to all readers please describe acronyms.**

The meaning of the IDAWEB acronym does not even seem to be explained even on the MeteoSwiss website (https://gate.meteoswiss.ch/idaweb/more.do). Essentially, it appears to simply be a name. To

reflect this, we will change the manuscript around line 205/206, where IDAWEB is first mentioned, to read *"were downloaded from the online data portal of MeteoSwiss"*. The link to the IDAWEB page can then be found via the reference which immediately follows. Reference to IDAWEB on line 339 will similarly be changed to "the data portal of MeteoSwiss".

In contrast, IMIS is explained by the SLF. It stands for the Swiss Intercantonal Measurement and Information System (IMIS). This acronym will therefore be explained in the revised version (line 340).

**p.15, l.345ff: Which parameters for the station was used by applying Jonas et al (2009)?**

To estimate snow density according to the model of Jonas et al. (2009), the appropriate monthly parameters (*b, a*) for the elevation band <1,400 m were used (the station elevation being 1,146 m). Since our station lies within "Region 1" of that study, the Region 1 offset of +7.6 kg/m$^3$ was also applied. The manuscript will be revised to make this clearer.

**p.15, l.354: I strongly disagree that these are actually direct snow measurements. This has to be corrected.**

Please see the earlier response (pg. 2 of this document). It was certainly not our intention to argue that these can be considered direct measurements (the contrary, in fact). We will rephrase the sentence to make this clearer. The new version will read *"neither of the "observed" SWE time-series, which are presented in Figure 5 (alongside their simulated counterparts from the final model), can be considered direct measurements."*

**Section 3.3: What is the temporal resolution of the streamflow data?**

The underlying resolution of the streamflow data is rather high, being per minute. However, for the purposes of this study, the hourly mean was calculated and used. The following sentence will be inserted at line 363 stating this: *"For this work, hourly mean flows were calculated."*

**p.16, l.362: Define 'rating curve construction'.**

To address this point, Line 362 will be modified to: "an empirical stage-discharge relationship (or rating curve)…"

Also, "rating curve construction" will be changed to "the development of the rating curve". (L366).

**Section 4: As you actually describe some methods also in Section 3, I would suggest to rename this chapter and relate it especially to your modelling, calibration and uncertainty estimation 'work' with WaSiM.**

Thank for this suggestion. Chapter 4 will be renamed to **"Numerical modelling".**

**p.16/l.365: without 'to'**

The original sentence makes sense in our opinion. But we will rephrase slight more concisely to *"...was selected as the foundation of our snow modelling approach".*

**p.19, Table 2: This table is not necessary and can be replaced by one sentence in the text.**

This will be done. The revised text will read *"...each pixel was binned into one of the quadrants of a contingency matrix (a: simulated "snow" and observed "snow", b: simulated "snow" but observed "no snow", c: simulated "no snow" but observed "snow", and d: simulated "no snow" and observed "no snow")".*

**p.20, l.486ff: Is there a reference for the chosen weighting? Or why did you choose the weighting of 60:40?**

There is no reference for the weighting. 60:40 was chosen so that both the spatial and temporal data had a major influence on the overall objective function, with the spatial patterns being slight dominant (the calibration based on spatial patterns being an important novelty of the study). A sentence will be added to the manuscript to make this clearer.

**p.22, l.537: 'were also developed': I suggest to write 'were also calculated'**

We agree that the term calculated is more appropriate here, and will make then change.

**Section 5: Please rethink your subchapter captions and a general merge with Section 6.**
**- p.26, l.603ff: For example, this point belong more to the discussion**

Yes, as mentioned in response to a similar point raised above, we feel that it could be better to have a combined Results and Discussion section. Subchapter headings will naturally be changed to accommodate this.

**- p.30, Figure 7a: Please explain more clearly what is compared and shown in this graph.**

This figure simply shows the temporal relationship between the hourly simulated "snowcover outflow" and hourly observed streamflow. To enable these quantities to be equated in mm, the distributed simulations were averaged across the Vallon de Nant sub-catchment, and the streamflow measurements in $m^3$/s were also divided by the catchment area (to give mm). The description in the text beginning at line 660 will be modified to try and make this clearer. The caption will also be lightly adjusted.

**References**

Cochand, M., Christe, P., Ornstein, P., & Hunkeler, D. (2019). Groundwater storage in high alpine catchments and its contribution to streamflow. Water Resources Research, 55(4), 2613-2630.

Warscher, M., Strasser, U., Kraller, G., Marke, T., Franz, H., & Kunstmann, H. (2013). Performance of complex snow cover descriptions in a distributed hydrological model system: A case study for the high Alpine terrain of the Berchtesgaden Alps. Water resources research, 49(5), 2619-2637.

---

## Referee Comment (RC2) · Anonymous Referee #2 · 6 Jan 2020

The authors present a multi-objective calibration strategy for a distributed snowcover and hydrology model WaSIM. This approach used Landsat 8 derived snow cover area as well as reconstructed snow water equivalent derived from observed snow depth and modelled and statistical density estimates to calibrate the snowmodel. This model was then applied to a basin in the Swiss Alps.

Although the authors have done a lot of work, I cannot recommend that this paper be published in its current form. The bulk of this manuscript relies upon the analysis of a temporally and spatially variable simulated snowcover in complex terrain. However, the snow processes used in this model do not reflect the current state of the art in cold

regions hydrology. That is, the modelling approach lacks the cutting-edge advances that a reader would expect to see in The Cryosphere for snowcover simulation in complex terrain. Further, there are many missing details regarding, for example, the initial conditions of critical components of the hydrological system, such as soils. Without these details it becomes difficult to fully evaluate the model and results.

Below I detail my specific concerns with the overall methodology.

The snow/rain phase determination relies on a calibrated air temperature threshold method. These air temperature threshold methods are known be less reliable than methods that consider atmospheric humidity or hydrometeor energy balances, e.g., Harder, et al (2013); Marks, et al (2013); Harpold, et al, (2017); Jennings, et al (2018). This is critically important for the mixed-phase conditions, which appears to likely occur at the lower elevation site as evinced by the mid-winter melts. If I understand correctly, wind undercatch is done in eqn1, and phase in eqn 3, however different thresholds are used, and the phase via eqn 3 does not inform the phase choice in eqn1 for undercatch. This seems counter the standard way of accounting for undercatch and phase. Phase determination should be done first, then the correct undercatch correction applied. The method presented here is unreferenced and without verification. Precipitation gauge undercatch has established correction factors (e.g., SPICE intercomparison project (Nitu, et al, 2012; Rasmussen, et al. 2012)) which should be used separately from phase determination.

A one-layer snowpack model is generally considered insufficient for deep mountain snowpacks, such as those in excess of 2000 mm presented here, e.g., Debeer, et al, (2010); Jennings, et al, (2018). Deep snowpacks develop vertical thermal gradients that are not represented with a one-layer model. Exchange layers are critical to accurately estimate outgoing longwave radiation (e.g., skin temperatures Essery, et al, (2015); Pomeroy, et al, (2016); Lafaysse, et al, (2017)), sensible heat fluxes, heat conduction, and ablation (i.e., cold content estimation with positive surface fluxes, Jennings, et al (2018)). The temperature at the base of the deep snowpack should not

directly inform the surface exchanges. I am not suggesting that the authors must use a many-layer snowpack model like SNOWPACK or Crocus. A few layers would be sufficient. I suspect this omission is why the energy balance and longwave calibration is required to correct for these missing processes.

During the energy balance model description, the text states that "...and [if] air temperature is favourable, melt can occur". This must not actually be what the authors intend. The entire purpose of an energy balance model is that air temperature does not determine melt, but rather that the surface energy balance does. If this is what is meant, then there is a misinterpretation of energy balance snowmelt models and the processes surrounding them. As it stands, this and the one-layer model are major limitations and deficiencies of the methodology.

I understand from the text that blowing snow was not included in the final simulation. However, they describe blowing snow as a simple precipitation correction based on Winstal's Sx parameter. This is incorrect. Rather, blowing snow mass is transported downwind through a fetch dependent turbulent advection-diffusion process due to entrainment of snow particles from the surface into the saltation layer once surface threshold shear stresses are surpassed. Significant mass loss is experienced with via blowing snow sublimation of in-transport particles due to their well-ventilated nature. The decrease in lee-side windspeeds causes entrained particles to deposit on the surface. I would encourage the authors to read the recent review by Mott, et al (2018). Note that this is a separate process from preferential deposition process as described by Lehning, et al (2008); Mott, et al, (2014).

Albedo decay is critical for energy balance snow cover modelling (Krinner, et al, 2018) and this is not described or cited. Albedo impacts the cold content of layers in the snowcover, in turn affecting outgoing longwave calculation and turbulent heat fluxes.

Incoming longwave radiation is impacted by cloudiness. There are many established parameterizations for this process based on T and RH that surpass pure calibration

with respect to transferability and physical basis. It seems that the approach herein is to calibrate the meteorological data via the correction factors? How is sub-canopy longwave handled?

The turbulent transfer approach is not detailed in the manuscript but looking at the model documentation suggests that there is no stability correction. Although issues with MO parameterizations causing excessive stability are known, none of these issues are touched upon nor the approach taken discussed (I assume it is a neutral stability assumption?).

Potential evapotranspiration is meaningless for this area due to there not being an infinite supply of water. Again, looking at the model code suggests that calculated PET is scaled by a soil water content. However, no soils information is provided, nor is there any indication as to what the initial conditions for the soils are. Further, why is PET even calculated for snowcovered surfaces?

Frozen soil infiltration is a critical component of cold regions runoff (Gray et al, 2001), however this process is not described, and it is not clear if it is used. This again relates to the initial conditions for the soil moisture and type. As well, it impacts the runoff values presented later.

There appears to be vegetation in this basin, however this is not particularly clear – Figure 1 is small. There is no description of canopy interception processes. Incepted snowfall in tree canopies can lead to high sublimation rates, which is not detailed or considered here.

Estimation of complex terrain wind velocities cannot simply be done by lapsing and spatially interpolating windspeeds. Windspeeds near crests increase, which is an important aspect of midwinter snowpack sublimation and blowing snow. Indeed, this can result in snow-free ridges in many snowpack models. Even the most rudimentary of wind speed interpolation models such as Liston, et al. (2006) would have better success in preserving the spatial pattern of wind velocity.

The calibration ranges as presented in Table 1 suggests a complete lack of physical basis and identifiability. For example, the air temperature lower-range is 0.000001C. The lower limit on "scmd" is equally nonsense – 0.0000001 kg/m^2 is measuring micrograms! As a modeller, I understand the desire to force the calibration scheme to behave. However, if this is to be presented as a physically based work, using identifiable parameters and values, such nonsense values cannot be used. Even a parameter like scmd, clearly a calibration parameter, should be constrained to reasonable values – values representing 0.00001% of a snowpack suggests that either this value is not relevant to the calibration or the model is unduly sensitive to tiny values, further undermining the claim on a physically based approach.

The radiation balance is not well described but it also raises further concerns. How is sub-canopy radiation calculated? Line 275 on page 11 suggests self-shadowing is considered, e.g., cosine slope correction of radiation. In complex terrain, horizon shadows (terrain shadowing other terrain) are a key process that results in late lying snowpacks and impacts the surface energy balance on the snow (e.g, net longwave, turbulent fluxes). There are many algorithms for this, such as Dozier and Frew (1990). I am sympathetic to not adding new process to a model being used, however I cannot help but expect that the longwave calibration factors are compensation for this process. Yet, there is no description of it in the text. Given all the other ignored processes that are key processes of high mountain hydrology, I worry the calibration process is compensating for major conceptual issues throughout the model. Without a discussion on this, it is difficult to fully quantify the magnitude of this.

The snowcover maps presented in Figure 3 concerns me. The authors first note an "automated calibration scheme", but then note a "bespoke" per-image NDSI threshold. As noted, this greatly limits the applicability of this approach. I am perplexed with some of the sub-figures in Fig 3. For example, 2017.11.14, 2018.02.02, 2018.03.22 have what seems to be entirely complete snowcover. Certainly, this is possible. But I am surprised that it occurs with such regularity and that no mass transport or sublimation

processes quickly cause there to be snow-free pixels. This will have enormous weight in the calibration scheme, especially if these are used with a vertical mass transport model. Further to the point about canopy interception, on line 327 p 14 the authors state "under dark forest". However, there does not seem to be any effort to distinguish between snow on the ground or snow in the canopy during NDSI calculation. I would expect this to impact calibration efforts. Were these areas masked out? This is a large uncertainty that does not seem to be addressed.

The SWE presented in Fig 5 suggests that the model does not perform adequately. During 2015 the GRY site peak SWE appears to be under-estimated by ~90%. This seems rather egregious if the goal is to provide water managers accurate and timely SWE estimates, as stated in the introduction. Similarly, 2016 has a ~2x error at COR and a 5x error at GRY. Certainly, models do not get every hindcast year right, but given the uncertainty in the reconstructed SWE as well as vast number of seemingly missing processes in the model, it is not clear that this error is down to uncertainty in reconstruction only. A 25% success rate on predicting SWE at the low elevation suggests significate deficit in the model. Given the mid-winter melt at this site, and that mid-winter melts are considered to become more likely (Musselman, et al 2017) it suggests this model is likely of limited use with future climate change scenarios.

I have concerns with the reconstructed SWE. Because the SWE is computed via a modelled density, this is essentially comparing the density model of SNOWPACK versus the density model of WaSIM + uncertainty in precipitation interpolation. The densification processes via compaction and settling are so complex that most models tend to have considerable uncertainty in simulated snowdepths. SNOWPACK is advanced in this regard due to its design as an avalanche hazard model, however it still has uncertainty in the densification processes. This is fine, however there is no discussion on this in the text. This comparison must consider this outside of an arbitrary (?) +-20% uncertainty bound. The lack of in-basin observations is a major limitation. Without any in-basin values, it is not at all clear that the reconstructed SWE values are representative of in-basin values.

I am missing what I would expect to be a standard description of the basin's instrumentation. This makes it difficult to follow the text. I am perplexed that the authors note that non-heated snowfall gauges were ignored. Non-heated snowfall gauges are the norm in every research basin except those with large mains power, which are a rarity. Perhaps it is just that the text is unclear, but this requires further elaboration.

I am further missing a complete description of the basin, of the observed climate, of the vegetation, soils, rivers, et cetera. I am missing a complete map of these in the main text. A better figure is present in the supplement, but this needs to be in the main text. Further, contour lines or a DEM would greatly aid in understanding the topography of the basin.

I am not convinced that the use of runoff without an overland routing model is appropriate. Mountain catchments do not instantly transmit surface waters to the rivers. There is soil infiltration, ground water recharge, and surface detention in ponds. I see WaSIM has an overland flow model for cell sizes <10 m – could this not have been used to route to the channel? Figure 7 b suggests, I guess, that the basin is flashy and that high melt rates are correlated with higher discharge at the weir, without considering any routing. Without any information on the soils, etc of the basin though, this is difficult to make much sense of or, importantly, transfer to other research sites. The note about integration into a 3D bedrock model suggest that the authors think there is groundwater recharge and groundwater contribution in the basin. This further suggests that no routing and overland flow is problematic.

The description of the interpolation is unclear; however, it reads as if no common elevation was used for the interpolation (e.g., Liston, et al 2006). That is, the stations' data were interpolated and then elevation corrected, thus incorporating an elevation dependence into the interpolation. This is not correct. This needs further description to ensure that this is not what is being done. In addition, IDW can cause significant

"wells" around stations as well as being poorly numerically conditioned. I would like the authors to describe why they chose IDW versus a spline, for example, which I understand to also be in WaSIM.

Lastly, the glacier model comes out of nowhere in the description and is not well detailed. Dynamic glacier models open up an entirely new set of uncertainties and difficulties that does not seem to be addressed here. Without details on how exactly this model was initialized (i.e., what is the initial state of the ice? Depth? Firn? Etc), it's impossible to evaluate this aspect of the manuscript.

In summary, the model methodology as presented herein is missing critical processes known to be important for cold regions hydrology in complex terrain, and those that are included appear to be potentially misapplied. There is also a substantial lack of details on initial conditions, parameters, and limited site descriptions.

Detailed comments

In general, I found this paper to be quite long and difficult to follow. The English tends to be conversational and needlessly verbose.

P2. L.13 "they" should this be "these data"?

P2. L.13 "exploited" word choice

P2. L.14 remove "-based"

P2. L.18 "practically the full range of possible snow cover conditions" has this really been shown?

P2. L.21 pt i) is wordy. Further, i) is well known, ii) seems to be a model formulation limitation, iii) does not seem to be a novel finding

P3. L.35 "Under established climatic conditions" unclear what this means

P3. L.44 Add Musselman 2017, Viviroli 2007

P3. L. 47 Add "changes to" patterns of liquid?

P3. L. 50 missing thesis statement, unclear what the paragraph trying to tell the reader

P3. L. Throughout, I would suggest avoiding conversational terms and expressions like "increasingly appreciate that"

P.4. L.81 I am surprised by the references here. Missing a lot of the detailed Alpine3D work, iSnobal, CRHM, CROCUS, etc done by Lehning, Mott, Marks, Pomeroy, Layfasse and Vionnet to name but a few, as well as obs+model hybrids like ASO.

P4. L. 83 "Firstly", then "Secondly". It's unclear what these are item enumerations are in reference to.

P4. L. 85 These are called sub-grid processes

P5. L.96 "certain applications" – such as?

P5. L. 97 "small" = what length scale ?

P5. L.104 Wayand, et al (2018) considered hundreds of images from Sentinal-2A and Landsat 8.

P5. L. 106 Standard in what context? There is a strong adoption of energy balance models to provide robust estimates outside of calibration periods, e.g., climate change predictions.

P5. L. 113 remove "much"

P5. L. 118 what does "explicitly physical fashion" mean?

P7. L. 167 the dynamic glacier model comes out of no where

P. 7. Study area. As stated above this is missing significant details such as veg cover, climatic information, etc.

P.8 Figure 1 caption: is $^{©}$ swisstopo really the correct way to cite these data? P.8 L.184

"Abundant precipitation", how much?

P.9 L. 205 What is IDAWEB? Citation is missing a date.

P.9. L. 214 "local station data" which stations?

P.9 L.215 How were these corrections actually done? What algorithms?

P.9 l.217 fix niVis citation, also unclear what this software really is

P10. L.239 "this parameter" what parameter does this refer to, specifically?

P11. L.274 I do not have Oke, 1987 handy, and I do not recall the temperature shading method used therein. Please detail.

P11. L.274 What is radc? Describe

P11. L.275 WaSIM comes out of nowhere, and without any detail. This model needs to be better described and introduced.

P12. L.300 Define "satisfactory"

P12. L. 303 Define EPSG

P14 L.325 Remove "The present study was no..."

P14 L. 327 "Dark forest", first mention of this, what are implications for modelling (e.g., interception, radiation)? Landsat tiles? Etc

P14 L.328 "Very pleasing" what does this mean? Is this numerically quanitifed or purely subjective?

P15 L.335 "fuller information", "interrelations" grammar

P15 L 339 "IMIS" write this out and fix n.d. citation

P15 L 346 "height" should be depth

P16 L 361 How often do these flows exceed the rating curve?

P16 L 365 "thorough review" There are a ton of fantastic energy balance snow models. It is really not clear to me that WaSIM is better, or what the evaluation criteria were here.

P16 L.367 I thought redistribution was not used?

P16 L376 What determined T_{trans}=1degC?

P16 L388 What determined 2W mˆ-2?

P17 L 389-391 As I described above, this is a fundamental problem with this model. Using an energy balance model with air temperature constraints on melt defeats the purpose of using an energy balance model.

P17 L398 the lw corrections for lw out are also compensating for a lack of skin temperature and the single bulk layer's temperature being used for lw out. Further, I imagine this will also compensate for the errors in turbulent fluxes as a result of the homogenous temperature.

P17 410 The purpose of frss is not clear. Why can only 1% of the snowpack slide?

P18 L453 "observed extent map had been produced" these are the landsat data, correct?

P20 Figure 4, personally I find this figure confusing. I.e., what are the % values in the figure? They are not shown in the caption.

P21 L510 is "coupled" really what you mean? I am not familiar with PEST, but I presume it is a driver that calls WaSIM and is not numerically coupled to WaSIM (as may be interpreted from the use of "coupled")

P21 L520 "predictions (this term is not used here in a future sense)" I do not understand this as predictions are always a future event/forecast.

P21 L526 What is a genuine future prediction?

P22 L530 I assume GENLINPRED is "Generalized linear prediction" or similar? This should be stated, and not just the internal acronym used.

P22 L541 "to account for undercatch", I understand this to mean eq1? I would state that explicitly or just remove it.

P22 L544 The glaciation treatment is vague. This should be better described. I.e., why is this glacier cover not described in the area section?

P23 L564 n.d. citations. Should also specify GDAL version

P23 L566 n.d. citation again (fix throughout)

P23 L566 remove "again"

P23 L569 "fuller example" grammar issue. Also unclear what "to this end" refers to. Suggest you remove this sentence.

P23 L571 Define "physically-based", as so far most of these methods are heavily calibrated and empirical

P23 L577 No ground water? Routing?

P24 Table 3 would benefit from plain text col headers

P26 L609 There are no standard numerical quantification of Figure 5 such as RMSE, bias, etc. This section is very qualitative.

P26 L610 "perfect simulator" This is not clear, seems very out of place. Of course, we know models not to be perfect and to have uncertainty.

P27 L634 The mid-winter melt in Figure 5 suggests otherwise?

P28 Figure 6 may benefit from having units changed to mm to match other figures' units

P38 L809 "Turning our attention to", conversational verbiage

P38 L811 Are not two of the data sets essentially the inverse of the other?

P39 L855 I don't think this was substantiated in the results that blowing snow is of negligible importance. Other authors have identified this process as critical in most mountain basins, including the Alps.

P39 S 6.6 This section misses a lot of the literature on the merits of not calibration snow models. Likely best to have this as part of the introduction to support the methodology taken in this paper?

References

DeBeer, C., Pomeroy, J. (2010). Simulation of the snowmelt runoff contributing area in a small alpine basin Hydrology and Earth System Sciences 14(7), 1205 1219.

Dozier, J., Frew, J. (1990). Rapid calculation of terrain parameters for radiation modeling from digital elevation data IEEE Transactions on Geoscience and Remote Sensing 28(5), 963-969.

Essery, R. (2015). A factorial snowpack model (FSM 1.0) Geoscientific Model Development 8(12), 3867 3876.

Gray, D., Toth, B., Zhao, L., Pomeroy, J., Granger, R. (2001). Estimating areal snowmelt infiltration into frozen soils Hydrological Processes 15(16), 3095-3111.

Harder, P., Pomeroy, J. (2013). Estimating precipitation phase using a psychrometric energy balance method Hydrological Processes 27(13), 1901-1914.

Harpold, A., Kaplan, M., Klos, P., Link, T., McNamara, J., Rajagopal, S., Schumer, R., Steele, C. (2017). Rain or snow: hydrologic processes, observations, prediction, and research needs Hydrology and Earth System Sciences 21(1), 1-22.

Jennings, K., Kittel, T., Molotch, N. (2018). Observations and simulations of the seasonal evolution of snowpack cold content and its relation to snowmelt and the snowpack energy budget The Cryosphere 12(5), 1595-1614.

Jennings, K., Winchell, T., Livneh, B., Molotch, N. (2018). Spatial variation of the rain–snow temperature threshold across the Northern Hemisphere Nature Communications 9(1), 1148. Krinner, G., Derksen, C., Essery, R., Flanner, M., Hagemann, S., Clark, M., Hall, A., Rott, H., Brutel-Vuilmet, C., Kim, H., Ménard, C., Mudryk, L., Thackeray, C., Wang, L., Arduini, G., Balsamo, G., Bartlett, P., Boike, J., Boone, A., Chéruy, F., Colin, J., Cuntz, M., Dai, Y., Decharme, B., Derry, J., Ducharne, A., Dutra, E., Fang, X., Fierz, C., Ghattas, J., Gusev, Y., Haverd, V., Kontu, A., Lafaysse, M., Law, R., Lawrence, D., Li, W., Marke, T., Marks, D., Nasonova, O., Nitta, T., Niwano, M., Pomeroy, J., Raleigh, M., Schaedler, G., Semenov, V., Smirnova, T., Stacke, T., Strasser, U., Svenson, S., Turkov, D., Wang, T., Wever, N., Yuan, H., Zhou, W. (2018). ESM-SnowMIP: Assessing models and quantifying snow-related climate feedbacks Geoscientific Model Development Discussions

Lafaysse, M., Cluzet, B., Dumont, M., Lejeune, Y., Vionnet, V., Morin, S. (2017). A multiphysical ensemble system of numerical snow modelling Cryosphere 11(3), 1173 1198.

Lehning, M., Löwe, H., Ryser, M., Raderschall, N. (2008). Inhomogeneous precipitation distribution and snow transport in steep terrain Water Resources Research 44(7)

Liston, G., Elder, K. (2006). A Meteorological Distribution System for High-Resolution Terrestrial Modeling (MicroMet) Journal of Hydrometeorology 7(2), 217-234.

Marks, D., Winstral, A., Reba, M., Pomeroy, J., Kumar, M. (2013). An evaluation of methods for determining during-storm precipitation phase and the rain/snow transition elevation at the surface in a mountain basin Advances in Water Resources 55(), 98 110.

Mott, R., Scipión, D., Schneebeli, M., Dawes, N., Berne, A., Lehning, M. (2014). Orographic effects on snow deposition patterns in mountainous terrain Journal of Geophysical Research: Atmospheres 119(3), 1419-1439.

Mott, R., Vionnet, V., Grünewald, T. (2018). The Seasonal Snow Cover Dynamics: Review on Wind-Driven Coupling Processes Frontiers in Earth Science 6(), 197.

Musselman, K., Clark, M., Liu, C., Ikeda, K., Rasmussen, R. (2017). Slower snowmelt in a warmer world Nature Climate Change 7(3), 214 219.

Nitu, R., Rasmussen, R., Baker, B., Lanzinger, E., Joe, P., Yang,D., Smith, C., Roulet, Y., Goodison, B., Liang, H., Sabatini, F.,Kochendorfer, J., Wolff, M., Hendrikx, J., Vuerich, E., Lanza,L., Aulamo, O., and Vuglinsky, V.: WMO intercomparison of instruments and methods for the measurement of solidprecipitation and snow on the ground: organization of theexperiment, WMO Technical Conference on meteorologicaland environmental instruments and methods of observations,Brussels, Belgium, 16–18 October 2012, 10 pp., availableat:http://www.wmo.int/pages/prog/www/IMOP/publications/IOM-109_TECO-2012/Session1/O1_01_Nitu_SPICE.pdf

Pomeroy, J., Essery, R., Helgason, W. (2016). Aerodynamic and Radiative Controls on the Snow Surface Temperature Journal of Hydrometeorology 17(8), 2175 2189.

Rasmussen, R., Baker, B., Kochendorfer, J., Meyers, T., Landolt, S., Fischer, A., Black, J., Thériault, J., Kucera, P., Gochis, D., Smith, C., Nitu, R., Hall, M., Ikeda, K., Gutmann, E. (2012). How Well Are We Measuring Snow: The NOAA/FAA/NCAR Winter Precipitation Test Bed Bulletin of the American Meteorological Society 93(6), 811-829.

Viviroli, D., Dürr, H., Messerli, B., Meybeck, M., Weingartner, R. (2007). Mountains of the world, water towers for humanity: Typology, mapping, and global significance Water Resources Research 43(7), 1-13.

Wayand, N., Marsh, C., Shea, J., Pomeroy, J. (2018). Globally scalable alpine snow metrics Remote Sensing of Environment 213(), 61-72.

---

## Author Comment (AC2) · 24 Jan 2020

Reviewer comments are in **blue**, author responses are in **black**.

The authors present a multi-objective calibration strategy for a distributed snow cover and hydrology model WaSIM. This approach used Landsat 8 derived snow cover area as well as reconstructed snow water equivalent derived from observed snow depth and modelled and statistical density estimates to calibrate the snowmodel. This model was then applied to a basin in the Swiss Alps.

Although the authors have done a lot of work, I cannot recommend that this paper be published in its current form. The bulk of this manuscript relies upon the analysis of a temporally and spatially variable simulated snowcover in complex terrain. However, the snow processes used in this model do not reflect the current state of the art in cold regions hydrology. That is, the modelling approach lacks the cutting-edge advances that a reader would expect to see in The Cryosphere for snowcover simulation in complex terrain. Further, there are many missing details regarding, for example, the initial conditions of critical components of the hydrological system, such as soils. Without these details it becomes difficult to fully evaluate the model and results.

We sincerely thank Reviewer 2 for reading our manuscript and for the extensive comments provided. However, we strongly but respectfully disagree with the overall assessment that was reached. The reviewer has clearly approached our manuscript from a very different perspective and philosophy to us; one that seems to place exclusive emphasis on the importance of the extremely detailed representation of all known physical processes that could conceivably influence the snowpack at very local scales. In addition, several important misconceptions appear to have been made. Consequently, whilst we respect the reviewer's extensive knowledge on physical snow processes and their numerical simulation, many of the points raised by the reviewer are not directly relevant to our study, as we demonstrate below.

We begin our response in very general terms. The complexity of any model must always be aligned with predictions that are made using it. As such, universally applying the most complex model possible can be counterproductive. Our study readily acknowledges that all types of models are uncertain and contain approximations. Even if one were to implement every single small-scale process mentioned by the reviewer, the model would still be a only simplification of reality. Avoiding calibration altogether could therefore be very dangerous. Moreover, if the calibration of such a model was undertaken, the the fitted parameters would still take up surrogate roles to compensate for structural deficiencies. Despite this, quantifying the uncertainties associated with somewhat simplified conceptualisations of the most relevant processes can provide a solid basis for decision making.

More specifically, our contribution acknowledges the considerable sources of uncertainty (e.g. in forcing datasets) that arise throughout snow modelling at catchment-scale in rugged and sparely instrumented alpine terrain. We propose a means by which the information content of this often patchy and uncertain input data can be maximised, and by which the residual uncertainty associated with predictions of interest made using the model can be quantified. In this sense, our approach is a general, universally applicable one rather that a detailed analysis of local parameters which would be difficult to generalise to other catchments. This in no way belittles the value and merit of looking into detailed and small processes as suggested by the reviewer; it is just an alternative (even complementary) perspective.

Several recurrent themes in the comments also suggest that the intention and novelty of our manuscript was significantly misunderstood by the reviewer in certain regards. Firstly, the reviewer seems to suggest that the novelty lies in the snow model itself, and that the code we selected is unsuitable: "the snow processes used in this model do not reflect the current state of the art in cold regions hydrology". Secondly, it seems to have assumed that we have presented a full hydrological model, despite it being explicitly stated otherwise in the manuscript (e.g. L576 of the original version): "there are many missing details regarding, for example, the initial conditions of critical components of the hydrological system, such as soils". Additionally, it seems to have been misinterpreted that we seek to identify parameter values that have physical meaning and therefore transferability outside this basin, which is not in fact out intention.

We would therefore like to immediately correct these apparent misunderstandings and make it clear that:

- We make no claim that the novelty of our study lies in the snow model itself; rather, the novelty lies in the approach to the multi-objective calibration of distributed snow simulations that is proposed, including associated predictive uncertainly and data worth analyses.

- The code we chose, which is widely used and cited, was a deliberate and appropriate choice for our study.

- We do not claim to present a full hydrological model. Hence, the comment above regarding missing information on critical components of the hydrological system is not relevant in this instance, and;

- We do not claim that the model parameters identified have physical meaning or transferability beyond this catchment.

That such misunderstandings have arisen highlights that we must to a better job in the revised version of the manuscript of communicating the intention of our paper. That said, we note that neither the comments of the Handling Editor nor Reviewer 1 aligned with those raised here.

Our more detailed response begins by discussing further these primary misconceptions. Then, the reviewer's specific comments are addressed. Whilst many of these have limited relevance in light of the significant misconceptions highlighted (and are therefore challenging to deal with), some do suggest useful changes which – once addressed – will improve the quality of the manuscript.

**General responses**

The main points that Reviewer 2 seems to have misconstrued are discussed sequentially below.

*The intention of the paper*

Contrary to the apparent interpretation of the reviewer, we make no claim whatsoever that the novelty of our paper lies in the snow model itself. After all, as the manuscript makes clear, the extension of WaSiM used in the present study has previously been described and exemplified by Warscher et al. (2013).

Equally, it is not the intention of this study to exhaustively simulate all snow processes that are known to occur in mountainous terrain. Nor does it seek to provide a basis for improving our collective understanding of small-scale snow processes. In these cases, the application of extremely sophisticated snow models of the type advocated by the reviewer in relatively small, extensively instrumented and surveyed research catchments would indeed represent an appropriate approach. In theory, under such circumstances, little or no calibration may be required to achieve excellent fits with measured data under such circumstances. Nevertheless, it would be important to keep in mind that any model, no matter how complex and sophisticated, is always a simplified representation of reality.

The situation far more commonly encountered in mountainous regions across the globe is that data, and especially meteorological data, are limited. Despite this, simulations are still required in these cases to inform environmental management decisions, and they should be as reliable as possible. It is such situations that the approach presented in our paper intends to target.

In particular, spatial fields of meteorological variables (which of course are required to force snow and hydrological simulations) are associated with substantial uncertainties in high and moderately-sized alpine catchments are associated with substantial uncertainties. As our contribution highlights, firstly, precipitation measurements are known to be biased towards underestimation by amounts that vary in time and space as well as with instrumental configurations. Secondly, due to the typically lower density of meteorological stations in high-evaluation terrain combined with strong spatial variability in meteorological phenomena driven by complex topography, there is uncertainty associated with the spatial interpolations.

These circumstances certainly prevail at the extremely rugged study site in question. The meteorological stations which provide the most complete and standardized data coverage, operated by the national agency (MeteoSwiss), are situated some distance from the study catchment and moreover are at much lower elevations. The stations located within the study catchment itself provide much more sporadic coverage (the temporal station data coverage was presented Supplementary Figure 2). Unfortunately, despite the efforts of SPICE and similar initiatives, which we do refer to in the paper, no perfect deterministic corrections exist that can be applied to immediately alleviate such uncertainties.

Thus, one is faced by uncertainties which are pervasive throughout the modelling process. Even in attempting to employ a model with a physically based core (to leverage some of their benefits), several important empirical parameters still need to be adjusted (or calibrated) to better represent the local conditions. Here, some such parameters relate to the representation of gravitation snow redistribution, for example, which very significantly influences the evolution of snow storage and melt patterns in extremely steep alpine terrain. Presently, this process cannot be represented across entire catchments using "calibration free", physically based algorithms.

**As such, the primary novel contribution of this study is the approach taken to the inverse problem, including the associated uncertainty and data worth quantification, rather than the particular mechanics of the snow model itself.** In other words, we propose a framework for the maximal exploitation of limited, patchy meteorological data in the context of high-resolution, distributed snow modelling for hydrological and ecological applications. Importantly, it would be straightforward and appropriate to apply our approach in similar catchments around the world, irrespective of the particular spatially-distributed snow model employed (i.e. our contribution is far from location or code specific).

Scale is also an important consideration in this regard. Snow processes occur across a range of spatial scales (e.g. Blöschl, 1999). Our goal here is not to reproduce small scale processes.

The reviewer's comments suggest that we were not successful in communicating this overall intention. The manuscript will be revised in order to rectify this.

*Suitability of chosen code*

Many comments seem rather critical of the suitability of the code chosen. However, **WaSiM, which is widely accepted and frequently used, was identified as having the key attributes that were required, given the aims of the investigation**. These capabilities include options for spatially interpolating meteorological station time-series with variable coverage, implementing an energy-balance approach, and representing gravitational snow redistribution, all whilst simultaneously making it possible to quantify glacial melt, liquid precipitation, and potential evapotranspiration (PET) on a fully consistent (i.e. spatially and temporally commensurate) basis. Considering glacial melt, liquid precipitation and PET in addition to snow is crucial to represent the boundary conditions that control the broader hydrological functioning or regime of steep, high-elevation alpine catchments.

As the reviewer is undoubtedly aware, some of the codes mentioned (such as SNOWPACK and Crocus) are only 1D dimensional snow models, and would therefore clearly have been unsuitable for the distributed modelling presented in our paper. Others do indeed facilitate a more comprehensive

description of physical snow processes operating within catchments, but come with their own compromises; CRHM, for instance, is not fully-distributed but instead operates at the scale of Hydrological Response Units (HRUs) (and therefore could be expected to "smooth out" the influence of rugged topography, for instance).

Evaluating the respective benefits and limitations of all available (distributed) snow models in great detail lay beyond the scope of our paper. In our view, there is no such thing as a perfect model, only choices which are suitable, or conversely less suitable, for given applications. Overall, we feel that the snow modelling approach we present does represent a substantial improvement over the far more simplistic approaches that continue to be applied in the context of hydrological modelling in mountain areas (as set out in the Introduction), even if it falls short of the extremely comprehensive and complex representation of physical processes advocated by Reviewer 2.

**Hydrological modelling**

We would like to correct the apparent misconception of the reviewer that our contribution includes a full hydrological model that generates simulations of stream discharge. This is not the case**. Our model purposefully does not include the subsequent movement of meltwater over the land surface, through the subsurface, or via the stream network**. The modelling work described in this paper ends with the generation of rasters describing the "arrival of liquid water at the land surface" (and corresponding rasters describing PET). We fail to see how the reviewer could have concluded otherwise.

More specifically, the discharge data presented are observations, and were introduced to demonstrate the "hydrological plausibility and consistency" of the snow simulations without restoring to a full hydrological simulation. This was achieved via a comparison with simulated daily catchment-averaged snowmelt. Therefore, details about soil properties and initial saturation states are not applicable. The manuscript will be adjusted where possible to make this even clearer.

As explained in the paper, the datasets generated as the output of this study will serve as boundary conditions to a physically-based, fully integrated flow simulator that accounts for 2D surface flows, 3D variably saturated subsurface flows, and bi-directional exchanges between these two domains. Such models are even more sophisticated than WaSiM for simulating hydrological processes in many regards, especially in geologically complex regions such as our study site (please see Thornton et al., 2018). This hydrological modelling work will be presented in a subsequent publication.

**Parameter identifiability and transferability**

As discussed above, in highly instrumented settings, it may be possible to reproduce snow observations using an energy balance snow model with limited or even no calibration (although this remains extremely challenging). Elsewhere, however, some degree of calibration is typically required. Essentially, this is because all models are simplifications of the complex reality, and so parameters must assume effective values (for instance, they cannot generally be parameterized directly using field measurements, even if the names of the measured quantities and model parameters are identical – not least due to issues of scale mismatch). Moreover, if a nullspace exists, different parameter values can provide equally good (or bad) fits to observed data – so called equifinality (Beven and Binley, 1992). In fact, one of the co-authors has recently contributed to a review article on these very topics (Guillaume et al., 2019), and so we are very much aware of the issues. A key conclusion of the review paper is that identifiably itself is not a problem; it is something that always arises unless we simplify the model structure to such a degree that no nullspace exists, which in most cases does not make sense. However, if multiple parameters provide an equally good fit to observations, this parameter space forms the basis for an uncertainty analysis.

For these reasons, **we make no claim that the parameter values identified have physical meaning such as would allow them to be transferred or regionalized to other sites (or indeed to facilitate**

**comparisons between sites)**. Instead, they must be (and are) treated as basin specific. In this sense, our methodology provides a means by which such parameters could be identified efficiently in other similarly data-limited catchments around the world, independent of where the particular choice selected lies along the "complexity spectrum". In our opinion, this is a point of view that is widely shared across the hydrological community.

As a final general point, we would like to discuss the length of the manuscript. We agree with both reviewers that the manuscript is already rather lengthy. Therefore, before submitting the revised version, we will carefully remove any superfluous material.

That said, we note that Reviewer 2 suggest that much additional technical information is added. This technical information would amount to repetitions of material from WaSiM's technical manual, which is already referred in the manuscript. Therefore, we do not see how it would be possible to give sufficient space could be afforded to the novel aspects of our contribution whilst also incorporating the additional details requested by Reviewer 1 and simultaneously keeping the manuscript to a manageable length.

**Responses to specific individual comments**

As a result of the general misconceptions outlined above, many of the specific comments do not warrant extensive responses in addition to what has already been said. Nevertheless, each is considered briefly below.

Below I detail my specific concerns with the overall methodology.

1. The snow/rain phase determination relies on a calibrated air temperature threshold method. These air temperature threshold methods are known be less reliable than methods that consider atmospheric humidity or hydrometeor energy balances, e.g., Harder, et al (2013); Marks, et al (2013); Harpold, et al, (2017); Jennings, et al (2018). This is critically important for the mixed-phase conditions, which appears to likely occur at the lower elevation site as evinced by the mid-winter melts. If I understand correctly, wind undercatch is done in eqn1, and phase in eqn 3, however different thresholds are used, and the phase via eqn 3 does not inform the phase choice in eqn1 for undercatch. This seems counter the standard way of accounting for undercatch and phase. Phase determination should be done first, then the correct undercatch correction applied. The method presented here is unreferenced and without verification. Precipitation gauge undercatch has established correction factors (e.g., SPICE intercomparison project (Nitu, et al, 2012; Rasmussen, et al. 2012)) which should be used separately from phase determination.

The air temperature threshold method was the only option for phase separation available in the code selected. Rather than assign a single fixed parameter *a priori*, however, we decided to allow this parameter to vary as a calibration target.

The reviewer's statement that two different thresholds were used in incorrect. As the manuscript makes clear, the "rain-snow temperature threshold", *rstt*, appears in both Eqns. 1 and 3, and takes the same value in each. In other words, the threshold parameters in both equations vary in tandem throughout each iteration of the calibration process. This was actually stated on L376 of the original manuscript.

Thus, a single parameter is identified as the phase transition threshold temperature, upon the basis of which the different undercatch corrections are applied in Eq 1. The same value is then defines the central point of the snow transition temperature range that is used to estimate the relative proportion of solid to liquid precipitation for each pixel and timestep in Eq 3.

In contrast to being unverified and unreferenced, this approach has been applied in many published studies using WaSiM (often without the additional refinement of allowing the threshold to vary through calibration), including those focusing specifically on rain-on-snow events (e.g. Rossler et al., 2014).

 A one-layer snowpack model is generally considered insufficient for deep mountain snowpacks, such as those in excess of 2000 mm presented here, e.g., Debeer, et al, (2010); Jennings, et al, (2018). Deep snowpacks develop vertical thermal gradients that are not represented with a one-layer model. Exchange layers are critical to accurately estimate outgoing longwave radiation (e.g., skin temperatures Essery, et al, (2015); Pomeroy, et al, (2016); Lafaysse, et al, (2017)), sensible heat fluxes, heat conduction, and ablation (i.e., cold content estimation with positive surface fluxes, Jennings, et al (2018)). The temperature at the base of the deep snowpack should not directly inform the surface exchanges. I am not suggesting that the authors must use a many-layer snowpack model like SNOWPACK or Crocus. A few layers would be sufficient. I suspect this omission is why the energy balance and longwave calibration is required to correct for these missing processes.

As Warscher et al. (2013, p. 2623) note, when one is concerned with the generation of snowmelt estimates at catchment/regional scales, processes occurring lower down in the snowpack are only of relevance insofar as they influence the energy balance of the surface.

In light of this, and given the overall of our study and data availability, we remain confident that our choice of (single layer EB) snow model achieves an appropriate balance between physical realism and practicality (e.g. with respect to runtimes). We consider the application of even this comparatively simple single layer energy balance scheme to represent a significant improvement upon the predominately temperature index schemes that continue to be heavily relied upon in contemporary hydrological modelling of alpine catchments.

The longwave calibration factors are indeed included in part to help mitigate against the face that the snow is not modelled using multiple layers, alongside compensating for potential errors in cloudiness and albedo. We are happy to state this openly.

As highlighted earlier, SNOWPACK and Crocus are not spatially distributed, so we agree that they would not be appropriate tools for this study. Alpine3D is another code that the author team has a good deal of experience with; this code was carefully considered for this study but was eventually not selected for reasons already outlined in our response to Reviewer 1.

3. During the energy balance model description, the text states that ". . .and [if] air temperature is favourable, melt can occur". This must not actually be what the authors intend. The entire purpose of an energy balance model is that air temperature does not determine melt, but rather that the surface energy balance does. If this is what is meant, then there is a misinterpretation of energy balance snowmelt models and the processes surrounding them. As it stands, this and the one-layer model are major limitations and deficiencies of the methodology.

In their explanation of the approach, Warscher et al. (2013, p. 2622) do indeed state that air temperature is used as a proxy to differentiate between melting and non-melting conditions. The reviewer may be correct in asserting that this was not explained as clearly as it might have been in our paper, however.

Warscher et al. (2013) actually state that where the air temperature ($T_a$) is > 273.16 K, then the snow surface temperature is assumed to be = 273.16 K. Then, if the energy balance is positive, melt occurs. If, on the other hand, $T_a$ < 273.16 K, then the snow surface temperature is adjusted until the energy balance residuals = 0 (Strasser 2008, 2011).

Using air temperature as a proxy in this way has generally proved stable (Weber, 2008), though it must be acknowledged that the air temperature could exceed the temperature threshold whilst the snow surface remains below it, causing some erroneous melt to be generated.

Schulla (2017) moreover adds that if the energy balance is positive, the available energy can be used to simulate either melt or sublimation. The wet blub temperature is used to decide which process (which of course have highly contrasting energy requirements) is to be favored.

Whilst we will seek to clarify the text somewhat in relation to these points, since it is already rather lengthy as is not a first application of this snow model set up, we do not feel it is either necessary not appropriate to regurgitate extensive technical descriptions of the underlying algorithms from the existing documentation. Citing the appropriate references, as we did, should surely be sufficient.

4. I understand from the text that blowing snow was not included in the final simulation. However, they describe blowing snow as a simple precipitation correction based on Winstal's Sx parameter. This is incorrect. Rather, blowing snow mass is transported downwind through a fetch dependent turbulent advection-diffusion process due to entrainment of snow particles from the surface into the saltation layer once surface threshold shear stresses are surpassed. Significant mass loss is experienced with via blowing snow sublimation of in-transport particles due to their well-ventilated nature. The decrease in lee-side windspeeds causes entrained particles to deposit on the surface. I would encourage the authors to read the recent review by Mott, et al (2018). Note that this is a separate process from preferential deposition process as described by Lehning, et al (2008); Mott, et al, (2014).

The reviewer is correct that blowing/wind redistribution of snow was not included in the final simulation.

We do not describe this process as a simple precipitation correction factor, but rather state that applying such a factor is one way in which the complex processes that occur in reality, which are adeptly described by the reviewer, can be represented in numerical models (including WaSiM). This will be made clearer in the final version of the manuscript.

In fact, I think we are in some agreement on this point; the fact that in our tests, the fits obtained with our "observations" actually became poorer when this simple wind redistribution algorithm was included would indeed suggest that it is too simple to reliably be applied in such terrain.

5. Albedo decay is critical for energy balance snow cover modelling (Krinner, et al, 2018) and this is not described or cited. Albedo impacts the cold content of layers in the snowcover, in turn affecting outgoing longwave calculation and turbulent heat fluxes.

Variability in snow albedo is accounted for. Minimum and maximum snow albedo values were set to 0.5 and 0.9 respectively. If possible, we will try to find a place in the revised version of the text to mention additional details regarding albedo.

As with the other parameterisations, readers can consult the WaSiM control file that was provided as part of the paper's supplementary materials.

As noted previously, the focus of our contribution (on calibration, uncertainty quantification and data worth analyses), and space limitations, meant that (in common with similar papers), it is not sensible to describe every single minor detail in the text.

6. Incoming longwave radiation is impacted by cloudiness. There are many established parameterizations for this process based on T and RH that surpass pure calibration with respect to transferability and physical basis. It seems that the approach herein is to calibrate the meteorological data via the correction factors? How is sub-canopy longwave handled?

Cloudiness/sunshine duration data are one of the meteorological inputs, and cloudiness is taken into account in the calculation of incoming longwave radiation via adjustment factors (which vary with cloudiness). See Schulla et al. (2017, p. 73). If space permits, this information will be added to the manuscript.

7. The turbulent transfer approach is not detailed in the manuscript but looking at the model documentation suggests that there is no stability correction. Although issues with MO parameterizations

causing excessive stability are known, none of these issues are touched upon nor the approach taken discussed (I assume it is a neutral stability assumption?).

The WaSiM manual provides few details on the approach taken in this regard. We will endeavour to check with the model developers and state the approach taken in the revised version. We note, however, that the overestimation of turbulent fluxes associated with a neutral assumption the reviewer alludes to must often be corrected for. This is another example of why it might be possible to apply sophisticated snow models without calibration in theory, but hardly ever is in practice.

8. Potential evapotranspiration is meaningless for this area due to there not being an infinite supply of water. Again, looking at the model code suggests that calculated PET is scaled by a soil water content. However, no soils information is provided, nor is there any indication as to what the initial conditions for the soils are. Further, why is PET even calculated for snowcovered surfaces?

It is likely that this comment has arisen due to the reviewer's misinterpretation that we present a full hydrological model. Potential evapotranspiration corresponds to the amount of energy available for evapotranspiration (i.e. the atmospheric demand), and is completely independent of water availability.

As was explained in the manuscript, PET was estimated in this paper to provide boundary conditions to a subsequent integrated hydrological model. It was important that this could be done on a consistent basis with the snow simulations (i.e. same grid and timestep), which enables the comparison presented in Figure 6 of the original manuscript.

In a full hydrological model, soil moisture would indeed be taken into account in addition to PET and vegetation properties in order to estimate actual evapotranspiration (AET). This explains why no information regarding soil types or initial conditions was provided.

9. Frozen soil infiltration is a critical component of cold regions runoff (Gray et al, 2001), however this process is not described, and it is not clear if it is used. This again relates to the initial conditions for the soil moisture and type. As well, it impacts the runoff values presented later.

Please see the above response. The runoff values presented are observed, not simulated. It is stated (around L665 in the original manuscript) that catchment states will certainly influence the rapidity and degree to which snowmelt is translated into streamflow, but that the simulation of these processes is not part of this particular paper. It would also not contribute in any way to the goal we are aiming at.

10. There appears to be vegetation in this basin, however this is not particularly clear – Figure 1 is small. There is no description of canopy interception processes. Incepted snowfall in tree canopies can lead to high sublimation rates, which is not detailed or considered here.

There is some forest present in the lower reaches of the catchment. As mentioned in an earlier response, a more detailed map of the catchment itself will be included in the revised version to enable such details (plus to topography) to be discerned more easily.

In the version of WaSiM used in this study, snow-canopy interactions, including interception and modified sub-canopy climatic conditions were missing. As explained around L416 in the original manuscript, this deficiency has now been addressed by Förster et al. (2018), but unfortunately the corresponding official release of WaSiM came at too late a stage in our project for these additional algorithms to be considered. The manuscript will be adjusted to make it more explicitly clear that these potentially important aspects are not considered by our model.

However, it is not of major concern for this particular study. Given the fairly limited forest cover, especially upstream of the gauging stations, this limitation will not compromise the generated meltwater dataset for the purposes of our subsequent hydrological modelling.

11. Estimation of complex terrain wind velocities cannot simply be done by lapsing and spatially interpolating windspeeds. Windspeeds near crests increase, which is an important aspect of midwinter snowpack sublimation and blowing snow. Indeed, this can result in snow-free ridges in many snowpack models. Even the most rudimentary of wind speed interpolation models such as Liston, et al. (2006) would have better success in preserving the spatial pattern of wind velocity.

Again, we chose the most appropriate approach that was available the code we chose to use. Specifically, we followed the advice of Schulla (2017, p.21) that elevation dependent regression (EDR) was suitable for "all elevation dependent types of input data like temperatures, vapour pressure, air humidity, wind speed. Recommended for basins with a substantial elevation range only".

Of course we accept that such an interpolation method such as that proposed by Liston and Elder (2006) have may improved matters. If our study had a somewhat different aim, it would have been crucial to interpolate wind speeds in a more sophisticated fashion. However, it fell beyond the scope of this study to implement another method that was not already available.

As with previous comments along similar lines, we do not feel that this is of central importance to our study, not least because – as the manuscript makes clear – wind blowing snow was not simulated in the final model anyway. Therefore, any inaccuracies in the interpolated wind field will predominantly "only" affect the correction of precipitation measurements, as well as the estimation of PET.

12. The calibration ranges as presented in Table 1 suggests a complete lack of physical basis and identifiability. For example, the air temperature lower-range is 0.000001C. The lower limit on "scmd" is equally nonsense – 0.0000001 kg/m^2 is measuring micrograms! As a modeller, I understand the desire to force the calibration scheme to behave. However, if this is to be presented as a physically based work, using identifiable parameters and values, such nonsense values cannot be used. Even a parameter like scmd, clearly a calibration parameter, should be constrained to reasonable values – values representing 0.00001% of a snowpack suggests that either this value is not relevant to the calibration or the model is unduly sensitive to tiny values, further undermining the claim on a physically based approach.

The parameters lower bounds mentioned took values of 0.000001 for numerical reasons. These parameters are log transformed, and so the lower bound of any parameter that one wishes to set to 0 must in fact be set to a marginally higher value. As such, these parameter bounds are most certainly not "nonsense", as the review intimates. The caption of Table 1 will be extended in the revised version of the manuscript to make this point clear. Moreover, as Table 1 indicates, the estimated value of the "scmd" parameter was 1.14, which is reasonable and not too dissimilar to the value of 1 employed by Warscher et al. (2013).

That said, and as explained in our general response above, we were not seeking to identify transferable parameters, merely those parameters which – within the physically meaningful or otherwise justifiable bounds specified – minimised degree of mismatch between multiple observations (and types of observations) given the forcing data available, model structure applied. Indeed, the parameterization depends on aspects such as the model timestep, grid resolution, and local topography. For example the fraction of snowpack which forms the slide is a fraction per timestep. Hence the rationale for including such parameters in the calibration.

In addition, we would like to emphasise that it is often good practice to allow a wide range of the parameter space to be explored, perhaps even including physically implausible parameters. This is because if the lowest objective function is obtained with physically unreasonable parameters, then this is a strong indication of structural deficiency in the model. In this sense, providing strict bounds might give a false sense of the structural adequacy of the model.

Lastly, in the most general sense, even employing empirical parameters within a physically based model is likely to be preferable in many cases to a purely statistical model.

13. The radiation balance is not well described but it also raises further concerns. How is sub-canopy radiation calculated? Line 275 on page 11 suggests self-shadowing is considered, e.g., cosine slope correction of radiation. In complex terrain, horizon shadows (terrain shadowing other terrain) are a key process that results in late lying snowpacks and impacts the surface energy balance on the snow (e.g, net longwave, turbulent fluxes). There are many algorithms for this, such as Dozier and Frew (1990). I am sympathetic to not adding new process to a model being used, however I cannot help but expect that the longwave calibration factors are compensation for this process. Yet, there is no description of it in the text. Given all the other ignored processes that are key processes of high mountain hydrology, I worry the calibration process is compensating for major conceptual issues throughout the model. Without a discussion on this, it is difficult to fully quantify the magnitude of this.

Yes, both the air temperature and radiation grids were both adjusted to account for topographic shading effects. We do not understand why the reviewer seems to suggest that this is not included.

We would especially contest the statement that "all the other ignored processes that are key processes of high mountain hydrology"; whilst these processes may be key to the evolution of the snow pack at very fine spatial scales, they are not key processes of high mountain hydrology more generally.

14. The snowcover maps presented in Figure 3 concerns me. The authors first note an "automated calibration scheme", but then note a "bespoke" per-image NDSI threshold. As noted, this greatly limits the applicability of this approach. I am perplexed with some of the sub-figures in Fig 3. For example, 2017.11.14, 2018.02.02, 2018.03.22 have what seems to be entirely complete snowcover. Certainly, this is possible. But I am surprised that it occurs with such regularity and that no mass transport or sublimation processes quickly cause there to be snow-free pixels. This will have enormous weight in the calibration scheme, especially if these are used with a vertical mass transport model. Further to the point about canopy interception, on line 327 p 14 the authors state "under dark forest". However, there does not seem to be any effort to distinguish between snow on the ground or snow in the canopy during NDSI calculation. I would expect this to impact calibration efforts. Were these areas masked out? This is a large uncertainty that does not seem to be addressed.

The need to employ "bespoke" thresholds does indeed pose a challenge to the generalisation of this approach, but was necessary to accurately map the observed snow extents.

Regarding the full winter snow cover in certain images, the reviewer may wish to consult mapped extents in relation to their corresponding true colour composite images in Supplementary Figure 4 (high resolution versions of which are provided in the online repository). From these images, the mapped snow cover extent on these days seems reasonable.

We acknowledge that mapping snow cover under dark forest canopies using NDSI can be difficult. However, we found that by carefully modifying the thresholds applied to individual NDSI maps in comparison with the true colour composite images enabled the extent of snow cover under forest canopies to be established fairly accurately. Again, please see Supplementary Figure 4. It should be noted that an open flat area in the lower part of the catchment was extremely helpful in inferring snow extents in the lower forested reaches.

More generally, as stated in the manuscript, for most images, the NDSI image histograms (the entire scenes, not just those clipped to the study area) contained two clear groups of pixels; a large mass of pixels with low values, corresponding to snow free regions, and a large mass of pixels with higher values, corresponding to snow covered pixels. Generally, even darker snow covered forest regions fell within the latter region, and the minima separating these regions was rather well defined. We additionally conducted some simple sensitivity analyses, exploring the extent to which the mapped extent changes with varying the snow/no-snow threshold in the region of this minima, and found the sensitivity to be low.

No masking out of forest areas was deemed necessary.

15. The SWE presented in Fig 5 suggests that the model does not perform adequately. During 2015 the GRY site peak SWE appears to be under-estimated by ~90%. This seems rather egregious if the goal is to provide water managers accurate and timely SWE estimates, as stated in the introduction. Similarly, 2016 has a ~2x error at COR and a 5x error at GRY. Certainly, models do not get every hindcast year right, but given the uncertainty in the reconstructed SWE as well as vast number of seemingly missing processes in the model, it is not clear that this error is down to uncertainty in reconstruction only. A 25% success rate on predicting SWE at the low elevation suggests significate deficit in the model. Given the mid-winter melt at this site, and that mid-winter melts are considered to become more likely (Musselman, et al 2017) it suggests this model is likely of limited use with future climate change scenarios.

As we suggested in the manuscript, these remaining mismatches are probably related to deficiencies in the meteorological data. It is important to note that these fits should not be compared with those obtained in previous studies where a weather station providing full time-series of many variables exists at, or extremely close to, the precise location at which snow observations are available and model-data comparisons are made. At GRY, only snow is measured, with the nearest meteorological measurements quite some distance away, whilst at COR, air temperature is measured but precipitation is not (actually only liquid precipitation is measured there – see our response to point 17 below for further information on that). Indeed, this distinction in the availability of local meteorological measurements is probably why the fits are so much better at COR than at GRY. Of course, the intermittent snow accumulation and melt sequences at the lower GRY amplify any errors in the spatially interpolated air temperature, for instance.

Contrary to what the review suggests, such information would prove useful to water managers, particularly with respect to highlighting the density of meteorological observations that would be required in such terrain to give highly reliable predictions of the evolution of SWE at discrete locations away from those stations.

With respect to the comment about the processes included, as we have discussed above, whilst some conceivable snow processes are not represented in the model, we feel strongly that there are few (if any) missing processes which are *important* given the scope and aims of this study – and that it is somewhat disingenuous for the reviewer to suggest that this is a vast number.

16. I have concerns with the reconstructed SWE. Because the SWE is computed via a modelled density, this is essentially comparing the density model of SNOWPACK versus the density model of WaSIM + uncertainty in precipitation interpolation. The densification processes via compaction and settling are so complex that most models tend to have considerable uncertainty in simulated snowdepths. SNOWPACK is advanced in this regard due to its design as an avalanche hazard model, however it still has uncertainty in the densification processes. This is fine, however there is no discussion on this in the text. This comparison must consider this outside of an arbitrary (?) +-20% uncertainty bound. The lack of in-basin observations is a major limitation. Without any in-basin values, it is not at all clear that the reconstructed SWE values are representative of in-basin values.

Contrary to the reviewer's suggestion, we do mention in the original manuscript that the reconstructed SWE values are themselves somewhat uncertain.

The uncertainties around the SNOWPACK derived SWE series is almost certainly rather small compared to the other uncertainties in the modelling processes (especially the precipitation interpolation). The IMIS network is composed of two different station types. The "wind" stations located close to ridges are measuring meteorological conditions related to wind and snow transport which are important for avalanche danger assessment. The "snow" station are located on flat terrain and are protected from the wind. From our own experience, SNOWPACK performs well at IMIS stations where the snow transport by the wind (erosion/deposition) is negligible, which is the case at the COR station. Estimating the uncertainty bound of such simulation is a complicated task, we are

however confident that SNOWPACK which is one of the most advanced (physically-based) and detailed snow model worldwide can provide good (even if not perfect) SWE estimates. We will add some discussion about the uncertainty in the main text.

The statistical density model of Jonas et al. (2009) which was relied upon at the GRY site is commonly used and highly cited. Being based on a large sample of field measurements, some information on the variability in the relationships between snow depth, density, and SWE in the Swiss Alps can be derived from that paper. The model also takes into account four variables: season, snow depth, elevation, and region. This will be made clearer in our revised manuscript.

Being situated slightly outside the study catchment, should the reconstructed station SWE time-series indeed not be representative of in basin conditions, this would not be highly relevant as the calibration scheme compares "observed" and simulated data at these station locations (in addition to the snow maps). Besides, as was already mentioned in the text, the fact that these stations are situated towards the upper and lower limits of the study catchment's elevational profile, in basin snow storage conditions can generally be expected to lie somewhere between these two cases (notwithstanding differences in slope, aspect etc.).

We explained in our response to Reviewer 1 why in basin measurements were not included in this study.

17. I am missing what I would expect to be a standard description of the basin's instrumentation. This makes it difficult to follow the text. I am perplexed that the authors note that non-heated snowfall gauges were ignored. Non-heated snowfall gauges are the norm in every research basin except those with large mains power, which are a rarity. Perhaps it is just that the text is unclear, but this requires further elaboration.

As stated in the manuscript, the in basin meteorological data in this study were provided by Michelon et al. (2017) and subsequent updates. Describing in detail the technical specifications of all the instruments at these stations was not considered particularly relevant to the understanding of the paper.

Supplementary Table 1 provides some summary information regarding all meteorological stations, whilst Supplementary Figure 2 illustrates the temporal coverage of the meteorological data.

If one includes non-heated gauges in this region, then winter precipitation is clearly enormously underestimated, as illustrated in the figure below. Even using specially calibrated correction factors, it would not have been possible to satisfactorily correct for such pronounced underestimation in a satisfactory fashion. It is therefore standard practice to remove such stations.

[Figure]

Figure 1. Hourly precipitation recorded by the non-heated rain gauge "Les Diableters SLF". Practically no winter precipitation is measured. In accordance with standard practice, precipitation data from such stations were removed from the study.

But it is certainly fair to state the challenge of power (solar was relied on at some) and communication between weather stations in such rugged and "closed" mountain terrain, along with harsh weather conditions (e.g. around 4 m snow depth in the upper reaches in January 2018) were largely responsible for the "gappy" nature of the in-basin time series.

18. I am further missing a complete description of the basin, of the observed climate, of the vegetation, soils, rivers, et cetera. I am missing a complete map of these in the main text. A better figure is present in the supplement, but this needs to be in the main text. Further, contour lines or a DEM would greatly aid in understanding the topography of the basin.

This description was deliberately kept minimal as we are not presenting a full hydrological model. We fully agree that a more detailed "closer up" figure of the catchment that conveys the topography should be included alongside the existing plates that show the wider meteorological network and the situation of the site within Switzerland. This will be included in the revised version of the manuscript.

19. I am not convinced that the use of runoff without an overland routing model is appropriate. Mountain catchments do not instantly transmit surface waters to the rivers. There is soil infiltration, ground water recharge, and surface detention in ponds. I see WaSIM has an overland flow model for cell sizes <10 m – could this not have been used to route to the channel? Figure 7 b suggests, I guess, that the basin is flashy and that high melt rates are correlated with higher discharge at the weir, without considering any routing. Without any information on the soils, etc of the basin though, this is difficult to make much sense of or, importantly, transfer to other research sites. The note about integration into a 3D bedrock model suggest that the authors think there is groundwater recharge and groundwater contribution in the basin. This further suggests that no routing and overland flow is problematic.

Please see our previous comments on this point – we are not presenting a full hydrological model, and so these comments do not apply to this particular paper.

20. The description of the interpolation is unclear; however, it reads as if no common elevation was used for the interpolation (e.g., Liston, et al 2006). That is, the stations' data were interpolated and then elevation corrected, thus incorporating an elevation dependence into the interpolation. This is not

correct. This needs further description to ensure that this is not what is being done. In addition, IDW can cause significant "wells" around stations as well as being poorly numerically conditioned. I would like the authors to describe why they chose IDW versus a spline, for example, which I understand to also be in WaSIM.

As stated around L250, different interpolation algorithms were applied for different variables. To account for the strong elevational dependence of many variables, Elevation Dependent Regression (EDR) was applied. For precipitation, to achieve a balance between the maintenance of spatial patterns and any elevational gradients, a combination of IDW and EDR. In all cases, advice provided in the WaSiM manual (Schulla, 2017) and also Viviroli et al. (2009) was followed example. Fuller descriptions of the interpolation algorithms used are provided in the code's documentation. Space did not permit this material to be reproduced in our paper.

That said, we are fully aware that generating reliable interpolations in such terrain is extremely difficult, which again leads in our view to an optimization approach being a good way of tackling the problem.

21. Lastly, the glacier model comes out of nowhere in the description and is not well detailed. Dynamic glacier models open up an entirely new set of uncertainties and difficulties that does not seem to be addressed here. Without details on how exactly this model was initialized (i.e., what is the initial state of the ice? Depth? Firn? Etc), it's impossible to evaluate this aspect of the manuscript.

Glacier melt was not the primary focus of this study. But as with PET, considering glacier melt using a simple dynamic glacier model was important for the broader work. The glacier model was therefore set up in a fairly standard fashion. Readers interested in this aspect are referred to the model's control file provided in the Supplementary Information. To put things in perspective, glaciers cover less than 3% of the catchment's surface area (as of 2013), hence glacier melt is relatively small.

22. In summary, the model methodology as presented herein is missing critical processes known to be important for cold regions hydrology in complex terrain, and those that are included appear to be potentially misapplied. There is also a substantial lack of details on initial conditions, parameters, and limited site descriptions.

Clearly, the reviewer did not really engage with the novel aspects of our manuscript, rather focussed instead on criticizing the fairly comprehensive and widely-used snow model that we chose to use. We thank the reviewer, however, for highlighting this as we clearly need to do a better job is highlighting the novelty of the approach.

**Detailed comments**

In general, I found this paper to be quite long and difficult to follow. The English tends to be conversational and needlessly verbose.

We agree that the paper is rather lengthy, although it is ironic that the reviewer wishes that a great many more details which are already in the code's documentation to be added. We also respectfully disagree that the language is not suitable, and note that both the original Handling Editor and Reviewer 1 considered the paper to be well written.

P2. L.13 "they" should this be "these data"?

This suggested modification will be made.

P2. L.13 "exploited" word choice

We propose changing this word choice to "realised".

P2. L.14 remove "-based"
This suggested modification will be made.

P2. L.18 "practically the full range of possible snow cover conditions" has this really been shown?

In terms of snow *coverage*, yes – from essentially none in the summer images to full in certain winter ones. We agree that this is separate to whether the full range of possible snow *storage* conditions have occurred over our simulation period.

P2. L.21 pt i) is wordy. Further, i) is well known, ii) seems to be a model formulation limitation, iii) does not seem to be a novel finding

We will change "the calibration process" to simply "calibration. i) may be well known in general, but here we are reporting the finding from this specific study. ii) this was the finding obtained given the data available and the specific code chosen, and so we consider it is appropriate to state it. It is important to note that the generic nature of our approach would enable similar conclusions to be reached irrespective of the data or model used in a particular study. iii) Again, this does not have to be a novel finding – the novelty is that it was obtained using our new calibration approach.

P3. L.35 "Under established climatic conditions" unclear what this means

This means under the climatic conditions that have prevailed over recent decades. In light of the extent and rapidly of ongoing changes in mountains catchments, it no longer seems appropriate to state "under present climatic conditions" in relation to the rest of the sentence.

P3. L.44 Add Musselman 2017, Viviroli 2007

Viviroli et al (2007) do not really discuss the application of hydrological models to make such predictions, which is the focus of this sentence. Likewise, whilst the results of Musselman (2017) do have implications for hydrological modelling under future climate, this paper concerns snow melt rates and so may not be the most appropriate to cite here.

P3. L. 47 Add "changes to" patterns of liquid?

Here we are focusing on the challenge of representing liquid precipitation fields even for historical events which were measured at gauges (i.e. going beyond only snow) when assessing the hydrological functioning of alpine catchments. But the reviewer is correct that future precipitation patterns may also evolve with climate change. There is much less certainly in the climate science related to even the direction of precipitation changes in mountainous regions, never mind their spatial patterns, that these aspects are some way off being considered in hydrological climate change impact assessments. We therefore propose to keep this sentence as is.

P3. L. 50 missing thesis statement, unclear what the paragraph trying to tell the reader

This paragraph (assuming the reviewer is referring to that starting on L51, as L50 is actually blank) is highlighting the limitations associated with calibrating hydrological models purely against the integrative measure of streamflow at the catchment outlet. This then leads on to previous attempts that have been made to include snow observations and their limitations, which our study addresses. We feel that this is important background information to include, and therefore propose keeping it as is.

P3. L. Throughout, I would suggest avoiding conversational terms and expressions like "increasingly appreciate that"

Thank you for this suggestion – we will try to remove such phrases throughout.

P.4. L.81 I am surprised by the references here. Missing a lot of the detailed Alpine3D work, iSnobal, CRHM, CROCUS, etc done by Lehning, Mott, Marks, Pomeroy, Layfasse and Vionnet to name but a few, as well as obs+model hybrids like ASO.

The references at this this point are referring to some of the latest hydrological modelling approaches and other studies that are being applied in such catchments more generally, whereas the authors the reviewer refers to deal more exclusively with snow related aspects.

P4. L. 83 "Firstly", then "Secondly". It's unclear what these are item enumerations are in reference to.

There enumerations are in reference to the limitations associated with the often prevailing methods. To make this clearer, we propose adding a sentence such as "There are several particular important limitations. Firstly,….".

P4. L. 85 These are called sub-grid processes

No, here we are talking specifically about spatially lumped or only partiallyy distributed hydrological models, which do not employ any concept of a grid at all.

P5. L.96 "certain applications" – such as?

Any applications where considering internal catchment snow patterns is important. This will be changed to read "many application in which internal catchment snow patterns are important" in the revised version.

P5. L. 97 "small" = what length scale ?

Well, any patches smaller than the 500 m pixel size of MODIS.

P5. L.104 Wayand, et al (2018) considered hundreds of images from Sentinal-2A and Landsat 8.

Thank you for drawing our attention to this relevant publication. We will change the text in the modified version at this point to read "with the notable exceptions of Hanzer et al. (2016) and Wayand et al. (2018).

P5. L. 106 Standard in what context? There is a strong adoption of energy balance models to provide robust estimates outside of calibration periods, e.g., climate change predictions.

These approaches remain standard in hydrological modelling for climate change impact assessments, especially in the European Alps but also elsewhere. The references provided at this point in the manuscript support this assertion.

P5. L. 113 remove "much"

This change will be implemented.

P5. L. 118 what does "explicitly physical fashion" mean?

This will be changed to "using physically-based algorithms".

P7. L. 167 the dynamic glacier model comes out of no where

The glacier model is briefly introduced at this point to highlight that this additional source of meltwater was also considered by our methodology, even if it is not the primary focus of the paper.

 As stated above this is missing significant details such as veg cover, climatic information, etc.

Some further details regarding the study catchment will be added at this point.

P.8 Figure 1 caption: is © swisstopo really the correct way to cite these data? P.8 L.184

This will be changed to "Source: Swiss Federal Office of Topography" (following Article 30, Geoinformation Ordnance).

"Abundant precipitation", how much?

Precipitation varies with elevation, but certainly > 1600 mm (annual average). This figure will be stated explicitly in the revised version of the manuscript.

P.9 L. 205 What is IDAWEB? Citation is missing a date.

Reviewer 1 also mentioned IDAWEB. This source will be renamed / explained in the revised version. The date 2019 will also be added.

P.9. L. 214 "local station data" which stations?

These are the ones operated by the University of Lausanne (see Supplementary Table 1). This information will be added in parentheses.

P.9 L.215 How were these corrections actually done? What algorithms?

Sorry, but L215 does not make reference to any corrections – we are not sure what the reviewer is referring to here.

P.9 l.217 fix niVis citation, also unclear what this software really is

niVis is a visualisation software developed by the SLF. In fact, the particular software that was used for generating these interactive plots for QC purposes is not really important to the understanding of the manuscript, so we propose removing this reference altogether in the revised version.

P10. L.239 "this parameter" what parameter does this refer to, specifically?

*rstt*. "*rstt*" will replace "this parameter" in the revised version.

P11. L.274 I do not have Oke, 1987 handy, and I do not recall the temperature shading method used therein. Please detail.

Essentially, this scheme is based on calculations of the sun's position and the terrain model. It is described thoroughly (in over 4 pages) in the WaSiM manual (Schulla, 2017, p. 49). It is not of central importance to this paper, and therefore does not in our view warrant further space.

P11. L.274 What is radc? Describe

*radc* is an empirical factor that is applied in the air temperature correction for topographic shading. As Table 1 reveals, this is applied as a multiplier of the original temperatures.

P11. L.275 WaSIM comes out of nowhere, and without any detail. This model needs to be better described and introduced.

We introduced WaSiM in this way as the focus of this paper is not intended to be the code itself, but rather the optimization framework that we propose (and to a lesser extent, the snow algorithms that were applied)

P12. L.300 Define "satisfactory"

By satisfactory, we mean classifications that gave good visual agreement with the True Colour Composite (TCC) images (upon which the classification thresholds were determined). This will be made clearer in the revised version of the manuscript.

P12. L. 303 Define EPSG

We do not feel that it is appropriate or necessary to define this – the geodetic parameter dataset is widely known as ESPG (see: http://www.epsg-registry.org/)

P14 L325 Remove "The present study was no. . ."

This phrase will be removed.

P14 L. 327 "Dark forest", first mention of this, what are implications for modelling (e.g., interception, radiation)? Landsat tiles? Etc

This section is concerned with the implications of "dark forest" for the snow mapping (from Landsat). Implications of such vegetation will be stated more clearly elsewhere in the manuscript.

P14 L.328 "Very pleasing" what does this mean? Is this numerically quanitifed or purely subjective?

This is a subjective visual assessment. The classification was done with reference to the TCCs, and so additional or ground truth data existed with which to independently evaluate the mapped extents. That the results of the observed extent mapping is very pleasing is hopefully evident in Supplementary Figure 4.

P15 L.335 "fuller information", "interrelations" grammar

This sentence will be modified to read "full information on the interrelation between snowpack density, depth, and SWE".

P15 L 339 "IMIS" write this out and fix n.d. citation

This will be addressed in the revised version (please see response to Reviewer 1).

P15 L 346 "height" should be depth

This will be addressed in the revised version.

P16 L 361 How often do these flows exceed the rating curve?

Flows informing the rating curve went up to ~2.5 m$^3$ s$^{-1}$. However, this is not really a crucial point to the understanding of the manuscript. Nevertheless, this information will be added in parenthesis to satisfy this comment.

P16 L 365 "thorough review" There are a ton of fantastic energy balance snow models. It is really not clear to me that WaSIM is better, or what the evaluation criteria were here.

This phrase will be changed to simply "review". Our need went beyond simply an energy balance snow model. Please see our more general response on page 3 of this document.

P16 L.367 I thought redistribution was not used?

Gravitational redistribution was. The word "gravitational" will therefore be inserted here in the revised version.

P16 L376 What determined $T_{trans}=1degC$?

Schulla (2107, p. 273) suggested that T_trans +/- 1 K was an optimal value for the nearby Thur basin in the Swiss Alps.

P16 L388 What determined $2W m^{-2}$?

Durot (1999) suggested that this is a robust value for mid-European Alpine sites. The appropriate reference will be inserted in the revised version.

P17 L 389-391 As I described above, this is a fundamental problem with this model. Using an energy balance model with air temperature constraints on melt defeats the purpose of using an energy balance model.

Please see our response to Point 3 above (p. 6 of this document).

P17 L398 the lw corrections for lw out are also compensating for a lack of skin temperature and the single bulk layer's temperature being used for lw out. Further, I imagine this will also compensate for the errors in turbulent fluxes as a result of the homogenous temperature.

N/A.

P17 410 The purpose of frss is not clear. Why can only 1% of the snowpack slide?

This parameter is the proportion of the snowpack that can slide per time-step, hence the relatively small values. This will be made clear in the revised version. Here, it was calibrated (the fixed 1% figure was applied by Warscher et al. (2013) who also used an hourly timestep).

P18 L453 "observed extent map had been produced" these are the landsat data, correct?

Yes, the maps derived from Lansdat data.

P20 Figure 4, personally I find this figure confusing. I.e., what are the % values in the figure? They are not shown in the caption.

This figure is just trying to illustrate visually the weighting scheme employed. The % figures represent the proportions of the total weight that are applied to each observation group and individual observation. Additional text will be added to the caption to better explain this.

P21 L510 is "coupled" really what you mean? I am not familiar with PEST, but I presume it is a driver that calls WaSIM and is not numerically coupled to WaSIM (as may be interpreted from the use of "coupled")

Thank you for this observation. We will change "coupled" to "linked".

P21 L520 "predictions (this term is not used here in a future sense)" I do not understand this as predictions are always a future event/forecast.

Here we use the term prediction in a more general sense, to mean "a model output of interest", which could be in the past or future. The manuscript will be adjusted to make this clearer.

P21 L526 What is a genuine future prediction?

Following on from the above response, "a model output of interest" that does correspond to an as yet unobserved future time period. Again, adjustments will be made to the revised version to clarify this.

P22 L530 I assume GENLINPRED is "Generalized linear prediction" or similar? This should be stated, and not just the internal acronym used.

We will add the necessary additional information here: "GENLINPRED, which enables linear predictive error analyses to be undertaken".

We feel it is appropriate to cite the specific name of the utility for potential future users (similarly to e.g. Schilling et al., 2014).

P22 L541 "to account for undercatch", I understand this to mean eq1? I would state that explicitly or just remove it.

Yes, exactly. We will add reference to Equation 1.

P22 L544 The glaciation treatment is vague. This should be better described. I.e., why is this glacier cover not described in the area section?

It was mentioned in L190 of the original manuscript that "several small glaciers persist at relatively low elevations in the north-facing upper reaches of both sub-catchments". The aerial proportion covered by glaciers will be added in the revised version. Regarding the glacial simulations, please refer to our earlier response (p. 12).

P23 L564 n.d. citations. Should also specify GDAL version

We will add that GDAL version 2.2.1 was used.

P23 L566 n.d. citation again (fix throughout)

This "n.d." citation emerged from following the journal referencing style in cases that references contained no date. However we will endeavour to assign appropriate dates.

P23 L566 remove "again"

This will be removed.

P23 L569 "fuller example" grammar issue. Also unclear what "to this end" refers to. Suggest you remove this sentence.

We will indeed remove this sentence.

P23 L571 Define "physically-based", as so far most of these methods are heavily calibrated and empirical

For example PET was estimated with Penman-Montieth.

P23 L577 No ground water? Routing?

Indeed, we do not present a full hydrological model. Please refer to our more general response on p.3 of this document.

P24 Table 3 would benefit from plain text col headers

We think it is important to keep the style in which $F_1$, $F_2$ and $F_3$ are referred to consistent throughout, therefore propose keeping this as it is.

P26 L609 There are no standard numerical quantification of Figure 5 such as RMSE, bias, etc. This section is very qualitative.

Given the spatial mismatch involved here (simulated pixel average vs observed point), in addition to potential uncertainties in the reconstructed SWE values, we did not feel that calculating such statistics, which are generally used to compare point time-series, would be very informative.

P26 L610 "perfect simulator" This is not clear, seems very out of place. Of course, we know models not to be perfect and to have uncertainty.

We propose removing the parenthetical text here: "(In reality, of course, no model is perfect; all are associated with some inherent biases and imperfections)".

P27 L634 The mid-winter melt in Figure 5 suggests otherwise?

We will add "except at low elevations."

P28 Figure 6 may benefit from having units changed to mm to match other figures' Units

We will make this suggested change to the figure in the revised version.

P38 L809 "Turning our attention to", conversational verbiage

This will be replaced with "Considering Figure 9(a)…."

P38 L811 Are not two of the data sets essentially the inverse of the other?

No, the different F-statistics are not the inverse of one another, but are certainly closely related to one another (hence the "redundancy" observed in this figure).

P39 L855 I don't think this was substantiated in the results that blowing snow is of negligible importance. Other authors have identified this process as critical in most mountain basins, including the Alps.

Indeed, our results do not demonstrate that wind redistribution is definitely of lower hydrological relevance than gravitational redistribution (only that including wind redistribution using the approach available seemed to lead to less good fits with our observations). Therefore, we will rephrase this sentence such that it reads more cautiously: "We suspect that blowing snow processes are of lesser hydrological importance than gravitational redistribution in our study catchment, although this may not by the case in other Alpine catchments".

P39 S 6.6 This section misses a lot of the literature on the merits of not calibration snow models. Likely best to have this as part of the introduction to support the methodology taken in this paper?

The reviewer is correct that we could have included an additional section in the introduction saying that under some circumstances (and this may even have merits with respect to performance under changed conditions). However, in addition to the common data limitations that we have already discussed, even where the snow component of a model may need to be hardly calibrated (as in cases where the meteorological data are very good and a sophisticated energy balance snow model is employed), the hydrological component almost certainly will be. In other words, in hydrology more generally, it is essentially a given that some calibration is required. Interestingly (at somewhat ironically given the general tone of the review), sophisticated snow models are often coupled with far simpler and more conceptual "boxes" which account for the remainder of hydrological cycle and ultimately streamflow generation, which have limited physical basis and are therefore heavily reliant on calibration.

Since this section discusses hydrological modelling approaches in mountain regions, we are happy with keeping the material where it is.

**References**

Beven, K., & Binley, A. (1992). The future of distributed models: model calibration and uncertainty prediction. Hydrological processes, 6(3), 279-298.

Blöschl, G. (1999). Scaling issues in snow hydrology. Hydrological processes, 13(14-15), 2149-2175.

Durot, K. (1999). Modelisation hydrologique distribuee du bassin versant nivopluvial de Sarennes. Validation de donnees d'entree et developpement d'un module de fonte nivale sous forêt. These de doctorat de l'Institut National Polytechnique de Grenoble, LTHE, Grenoble.

Förster, K., Garvelmann, J., Meißl, G., & Strasser, U. (2018). Modelling forest snow processes with a new version of WaSiM. Hydrological sciences journal, 63(10), 1540-1557.

Guillaume, J. H., Jakeman, J. D., Marsili-Libelli, S., Asher, M., Brunner, P., Croke, B., ... & Stigter, J. D. (2019). Introductory overview of identifiability analysis: A guide to evaluating whether you have the right type of data for your modeling purpose. Environmental Modelling & Software.

Jonas, T., Marty, C., & Magnusson, J. (2009). Estimating the snow water equivalent from snow depth measurements in the Swiss Alps. Journal of Hydrology, 378(1-2), 161-167.

Liston, G. E., & Elder, K. (2006). A distributed snow-evolution modeling system (SnowModel). Journal of Hydrometeorology, 7(6), 1259-1276.

Michelon, A., Schaefli, B., Ceperley, N. C. and Beria, H.: Weather dataset from Vallon de Nant, Switzerland, until July 2017, , doi:10.5281/zenodo.1042472, 2017.

Rössler, O. K., Froidevaux, P. A., Börst, U., Rickli, R., Romppainen-Martius, O., & Weingartner, R. (2014). Retrospective analysis of a nonforecasted rain-on-snow flood in the Alps–a matter of model limitations or unpredictable nature?. Hydrology and earth system sciences, 18(6), 2265-2285.

Schulla, J.: WaSiM (Water balance Simulation Model) Model Description, Zürich. [online] Available from: http://www.wasim.ch/downloads/doku/wasim/wasim_2015_en.pdf, 2017.

Thornton, J. M., Mariethoz, G. and Brunner, P.: A 3D geological model of a structurally complex Alpine region as a basis for interdisciplinary research, Sci. Data, 5, doi:10.1038/sdata.2018.238, 2018.

Warscher, M., Strasser, U., Kraller, G., Marke, T., Franz, H. and Kunstmann, H.: Performance of complex snow cover descriptions in a distributed hydrological model system: A case study for the high Alpine terrain of the Berchtesgaden Alps, 1210 Water Resour. Res., 49(5), 2619–2637, doi:10.1002/wrcr.20219, 2013.

Weber, M. (2008). Mikrometeorologische Prozesse bei der Ablation eines Alpengletschers: Abhandlungen/Bayerische Akademie der Wissenschaften, Mathematisch-Naturwissenschaftliche Klasse.